# Recurrent connectivity supports higher-level visual and semantic object representations in the brain

Jacqueline von Seth [1], Victoria I. Nicholls[2], Lorraine K. Tyler[2,3] & Alex Clarke [2✉]

Visual object recognition has been traditionally conceptualised as a predominantly feedforward process through the ventral visual pathway. While feedforward artificial neural networks (ANNs) can achieve human-level classification on some image-labelling tasks, it's unclear whether computational models of vision alone can accurately capture the evolving spatiotemporal neural dynamics. Here, we probe these dynamics using a combination of representational similarity and connectivity analyses of fMRI and MEG data recorded during the recognition of familiar, unambiguous objects. Modelling the visual and semantic properties of our stimuli using an artificial neural network as well as a semantic feature model, we find that unique aspects of the neural architecture and connectivity dynamics relate to visual and semantic object properties. Critically, we show that recurrent processing between the anterior and posterior ventral temporal cortex relates to higher-level visual properties prior to semantic object properties, in addition to semantic-related feedback from the frontal lobe to the ventral temporal lobe between 250 and 500 ms after stimulus onset. These results demonstrate the distinct contributions made by semantic object properties in explaining neural activity and connectivity, highlighting it as a core part of object recognition not fully accounted for by current biologically inspired neural networks.

[1] MRC Cognition and Brain Sciences Unit, University of Cambridge, Cambridge, UK. [2] Department of Psychology, University of Cambridge, Cambridge, UK. [3] Cambridge Centre for Ageing and Neuroscience (Cam-CAN), University of Cambridge and MRC Cognition and Brain Sciences Unit, Cambridge, UK. ✉email: ac584@cam.ac.uk

Successful visual object recognition enables us to see and understand the world around us. It is well established that the ventral visual pathway (VVP) critically supports this process[1–4], with activity within the first few hundred milliseconds supporting the recognition of the visual input[5–15]. Our increasingly detailed computational, cognitive and translational accounts of the spatiotemporal processes that underlie recognition allow us to form an increasingly specific understanding of what kinds of information may be linked to the evolving object representations, and how they are spatially and temporally distributed. Visual object recognition intrinsically involves accessing semantic knowledge about objects, highlighting the necessity to consider how visual and semantic properties account for the neural dynamics during recognition. While prior studies have shed some light on how semantic object processing relates to, and is distinct from, processing the physical visual attributes of the image[5,10–12,15,16], in this study we look to probe the evolving nature of object representations in a new domain, asking how dynamic connectivity patterns relate to the visual and semantic aspects of objects. Using a representational connectivity approach[17] will allow us to reveal the core object properties that feedforward and feedback connectivity relate to, helping to shape cognitive accounts of how we understand the meaning of what we see.

Traditional models argue that visual object recognition is underpinned by a largely feedforward process along the ventral visual pathway[2,14,18,19], while inter-regional feedback is reserved for more complex situations such as occlusion or visually degraded images[18,20–23], meaning it might not be initiated for unambiguous familiar images. However, it has also been argued that object recognition generally involves inter-regional recurrent connectivity[23–31]. Yet the functional consequence of feedforward and feedback signals, in terms of what kinds of representational details they support, is largely unspecified. One argument put forward to explain when recurrent processes might be more involved, is that when detailed semantic representations are required (such as knowing something is a tiger compared to knowing it is a living thing), long-range recurrent connectivity between anterior and posterior ventral temporal cortex increases[25]. However, it remains unclear exactly which aspects of visual and semantic object information relate to feedforward and feedback signals when recognising unambiguous familiar objects. To address this, we use a recently developed method for model-based representational connectivity analysis (RCA)[17,32], which builds on prior examples of understanding connectivity using representational similarity approaches for fMRI[33–36]. In the case of time-resolved data, RCA can show how activity in one region contributes to information-specific activity in another region at a later point in time, with the potential to reveal how feedforward and feedback signals relate to visual and semantic information. This follows the recent development of multivariate informational connectivity approaches[33,37–39], which seek to provide information about the timing and direction of between-region connectivity, alongside pointing towards the representational content connectivity can support. To infer timing and direction, RCA uses similar principles to Granger Causality, in that it evaluates whether past information in one region can help explain the current representational patterns in another region. While this kind of approach has been used in a number of studies to evaluate if there is shared information between regions over time (e.g.[29,40,41]), the RCA approach we employ here goes one step further by specifying the precise nature of that shared information[17,32]—namely, is connectivity helping shape visual or semantic representations.

A further important issue concerns whether representations can be sufficiently captured by models of visual information alone. One of the dominant contemporary approaches to understanding the computational properties of the VVP, is to assess the relationship between convolutional neural network models of vision and the brain. A range of biologically inspired neural networks with different architectures have been tested against human neuroimaging data, with locally recurrent models generally outperforming feedforward models. While artificial and deep neural networks have shown an impressive ability to capture something about the neural properties of the human VVP[9,12,42–44], it remains in question whether such models have the potential to fully capture object recognition, without considering an additional, qualitatively different, form of information —that of semantics. While ANNs have shown an impressive ability to distinguish between different objects, they have done so by discerning a label for an image. However, this cannot capture how different semantic concepts relate to one another, such as apples and bananas having some shared meaning. If the goal of visual object recognition is to understand what the object being perceived *is*, then this requires semantic memory. An essential aspect of semantic knowledge is that it is relational. Semantic distances have long been thought to capture the underlying organisation of semantic memory e.g. [45], which can be used to understand that bananas and apples share something in common despite being perceptually very different. While numerous studies have shown effects in the VVP attributed to semantics during object recognition[15,42,46–53], questions remain over whether these effects are attributable to object semantics, or if they actually relate to higher-level visual object properties.

It is undeniable that accessing semantic memory for an object is a clearly dissociable form of information from visual processing, as demonstrated through specific deficits[54,55] and studies controlling visual input while manipulating semantics (e.g.[25,56–58]). This highlights the importance of understanding the relationship between visual and semantic processes and the timing of such informational types. Particularly, we might expect that connectivity patterns may differentially relate to different informational (visual versus semantic) properties. In this regard, image-based neural network models offer a useful tool to investigate visual processes, and allow us to determine what, if anything, models of semantics can explain beyond the current state-of-the-art neural network models.

Here, we address two overarching issues relating to, first, the role of feedforward and feedback signals during objects recognition, and second the relationship between visual and semantic object properties. To achieve this, we use both fMRI and MEG with representational similarity analysis (RSA)[35,59] and MEG-RCA applied to the recognition of visual objects from a large variety of object categories. Building across complementary analyses we (1) use fMRI searchlight RSA to reveal the cortical architecture related to semantic object properties and how they relate to those explained by a computational model of vision; (2) explore the relative temporal dynamics and connectivity of visual and semantic measures at the level of MEG sensor arrays using both RSA and RCA; and (3) examine the spatio-temporal distribution of semantic effects, beyond those explained by a computation model of vision, using searchlight RSA of MEG source localised neural patterns. Finally we (4) test how connectivity within the resulting network relates to semantics and visual object properties. Together these analyses illustrate how visual and semantic processes relate to one another over time and space during object recognition. Importantly, these analyses shed light on the visual and semantic representations that might be supported by feedforward and feedback signals across regions along the VVP.

## Results

Utilising the CORnet-S artificial neural network (ANN)[60] of visual processing and a cognitive model of semantic object knowledge[61], we probed the spatial, temporal and connectivity

characteristics of visual object recognition using RSA in fMRI and MEG, and representational connectivity analysis (RCA) of MEG signals. In both fMRI and MEG datasets, participants viewed and named a range of visual images including different animals, foods, vegetables, fruits, vehicles, musical instruments, tools, clothing, and other common household objects (131 items in fMRI, 302 items in MEG). Using the responses to these individual concepts, we could then test whether brain-based similarity between the items related to our visual and semantic measures (see methods).

We used a pre-trained version of CORnet-S to obtain visual measures of our stimuli (extracted from THINGSvision[62]). CORnet-S is composed of different processing units which are conceptualised as capturing the visual areas V1, V2, V4, and IT of the non-human primate brain[60]. CORnet-S has locally recurrent processing within each area (with no between area feedback), and is amongst the best performing models in predicting neural responses recorded in macaque IT, and performs well in capturing human behavioural similarity judgements[28,60]. Nodal activations for each area of CORnet-S were extracted from the area's convolutional layer. From these, we calculated representational dissimilarity matrices (RDMs) that quantify how similar or different the activations of the layer were between all the images.

Modelling the semantics of objects requires an approach that defines semantic similarities and differences between individual concepts. This was achieved by modelling the relationships between concepts according to the semantic features associated with each individual concept (e.g. *has legs*, *has stripes*, *lives in India*, *is dangerous* for a tiger). Utilising a large-scale property-norming study[63], the similarity between object concepts can be calculated based on the amount of features two concepts share, resulting in a semantic feature RDM. Similarity therefore captures both superordinate category structure (as objects from the same category will have many overlapping features) and additional within-category individuation (as each member of a category will have a unique set of features). These RDMs from CORnet-S and semantic features were then tested against RDMs derived from fMRI and MEG (Fig. 1).

**Semantic brain networks beyond ANNs: fMRI RSA searchlight.** We first looked to establish which regions were associated with the layers of the CORnet ANN, and if a model of semantics could uniquely explain voxel patterns over and above CORnet. The four network layers of CORnet all showed significant RSA model correlations to the VVP including occipital and posterior temporal lobes. This included primary visual cortex, lateral occipital cortex, and the posterior fusiform (Fig. 2a; Supplementary Table 1). Many of these regions also significantly related to the semantic feature model, which showed additional significant model correlations with more anterior regions of the fusiform and parahippocampal gyrus, and ventral and medial aspects of the anterior temporal lobe (Fig. 2a, b).

The semantic feature model could uniquely explain voxel pattern similarity over and above that which could be explained by CORnet, in bilateral pVTC, lateral occipital cortex, posterior medial cortex and medial ATL (Fig. 2c). While there was some spatial overlap with CORnet models in pVTC, only the semantic model significantly related to voxel patterns in anterior temporal, posterior medial and anterior parts of the posterior ventral temporal cortex. This pattern is similar to our other analyses of semantics using the same dataset that characterised visual properties using different ANNs (AlexNet and HMax) and controlled for different aspects of semantic similarity structure[42,48]. Our fMRI analysis here provides a comparison to our MEG source localised RSA effects whilst also testing if semantic feature effects remain after controlling for all layers of CORnet-S.

**Distinct dynamics of low and higher-level visual and semantic processing: MEG Sensor-level RSA.** We next examined the temporal dynamics of object recognition with RSA of MEG sensor-level signals in relation to the same models of visual and semantic properties (Fig. 3; Supplementary Table 2). Looking at the unique effects of each CORnet model layer and the semantic model (see Supplementary Fig. 1 for non-partial correlation results), all three of the lowest layers of CORnet showed rapid but transient significant model correlations peaking near 100 ms (CORnet-V1, cluster p = 0.046; cornet-V2, cluster p = 0.011; CORnet-V4, cluster p = 0.003), with secondary effects of CORnet-V1 near 400 ms (cluster p = 0.016). The higher-level visual information captured through CORnet-IT showed later model correlation from around 130 ms, peaking near 250 ms, and remaining significant until approximately 600 ms (cluster p < 0.0001). The semantic model showed a distinct temporal morphology compared to the visual layers, becoming significant from around 250 ms with peaks near 350 ms and 600 ms (cluster1 p = 0.001; cluster2 p = 0.023; cluster3 p = 0.0002).

An analysis of the latency of the peak RSA effect sizes was conducted by calculating the 95% confidence intervals of the differences in RSA peak latencies between model RDMs (Fig. 3b). This further revealed that the semantic feature model peaked later compared to the low and mid-level CORnet layers (V1 95% CIs of peak difference [172–288 ms], V2 95% CIs [146–262 ms] and V4 95% CIs [174–292 ms]), indicated by the 95% confidence intervals not including zero, but not CORnet-IT (95% CIs [-4–124 ms]), while CORnet-IT peaked later compared to the low and mid-level CORnet layers (V1 95% CIs [156–198 ms], V2 95% CIs [128–178 ms] and V4 95% Cis [156–202 ms]). This suggests that, in addition to the spatial hierarchy seen in fMRI, we see a temporal evolution of object properties from low, to high-level visual properties, before semantic effects emerge later in time. To establish the generalisability of these effects across different biologically plausible ANNs, the analysis was repeated using CORnet-RT[64], Alexnet[65], Resnet50[66] and VGG19[67], with all analyses showing semantic effects in this time range beyond that explained by the ANNs (see Supplementary Fig. 2, Supplementary Table 3).

**MEG sensor-level connectivity of visual and semantic effects.** While these results might point to parallel spatial and temporal progressions of the information represented in hierarchically increasing feature-complexity, it reveals little of how regional interactivity might underly these effects. To address this, we utilised RCA of the sensor data to establish whether feedforward and/or feedback connectivity was related to the different visual and semantic properties (Fig. 4a). To do this, we split the MEG sensors into posterior and anterior regions and assessed RCA between them to provide a global measure of feedforward and feedback informational connectivity, as developed in Karimi-Rouzbahani et al.[32].

The analysis aims to test what influence past representational similarity in a source region (e.g. posterior sensors) has on future RSA effects of semantics in a target region (e.g. anterior sensors). For example, if there is feedforward connectivity whereby patterns in posterior regions influence patterns in anterior regions that relate to semantics, then those patterns in posterior regions should help explain variance in the anterior patterns. We can assess this by calculating two region-specific, time-lagged RSA time-courses and finding the difference. First, we calculate RSA effects in the target region for a model, and second we calculate RSA effects in the target region for a model while controlling for past neural similarity effects in the source region. If the neural patterns in the source region help explain something

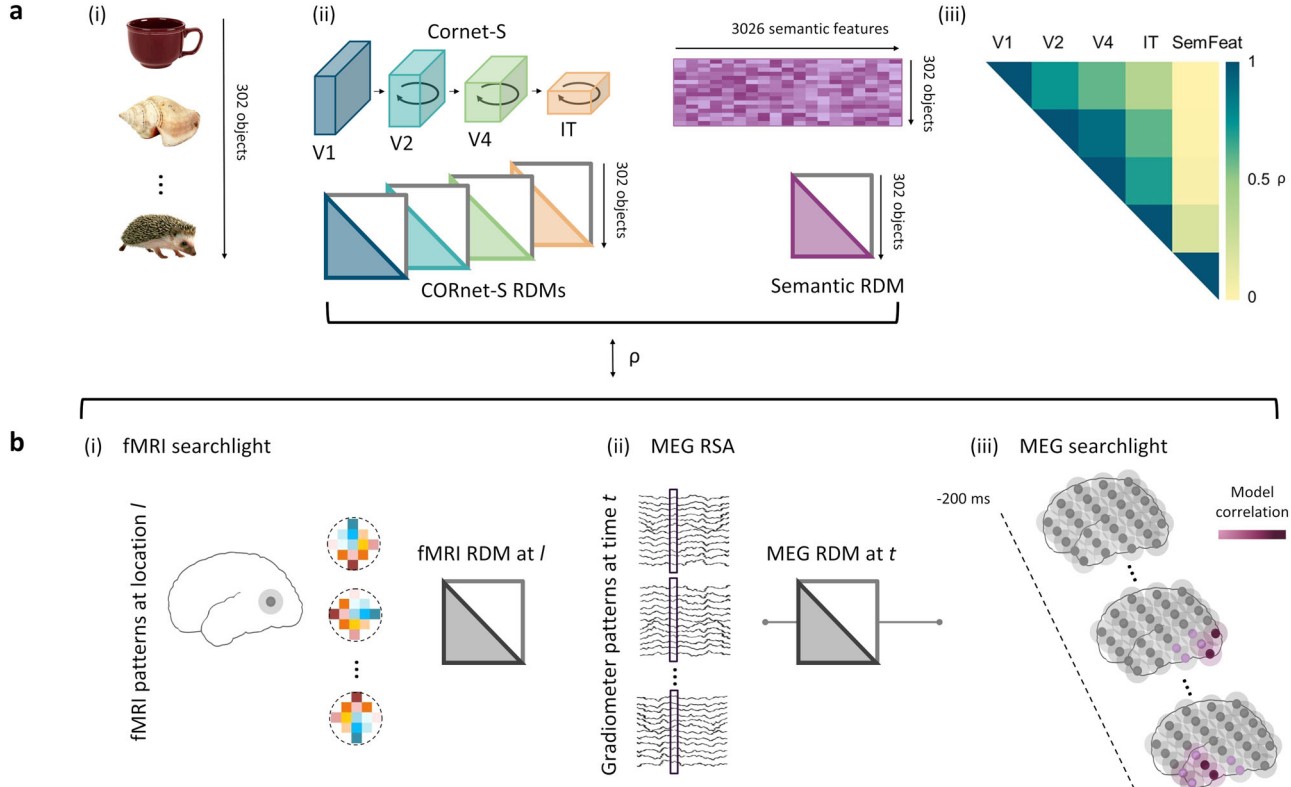

**Fig. 1 Schematic illustration of model and neural RDM construction for representational similarity and representational connectivity analyses.**
**a** Construction of model RDMs. (i) 302 objects including animate and inanimate, natural and man-made objects were shown to participants and used to construct model RDMs. (ii) Visual RDMs are created from pairwise comparisons of nodal activations extracted from 4 layers of the CORnet-S ANN for each object and vectorised. A semantic model RDM is created based on data from a large property norming study which generated 3026 features, with the RDM defined by the overlap in features between concepts, and vectorised. (iii) Pairwise Spearman's correlations between visual and semantic feature models show a high degree of correlation between the visual RDMs, graded by distance, but limited correlation between visual and semantic models.
**b** Construction of vectorised model RDMs from MEG and fMRI data (note, MEG RDMs based on 302 objects and fMRI RDMs based on 131 objects). (i) fMRI searchlight RDMs reflect the similarity between voxel patterns for each of the objects, and each searchlight location across the brain. (ii) For the sensor-level MEG RSA analysis, MEG RDMs are created from object-specific spatio-temporal patterns for each time-point extracted from MEG sensors. (iii) Temporally resolved MEG RSA searchlight analysis is conducted for the semantic model using source localised MEG patterns. Vertices are illustrated with grey dots with shaded searchlight spheres, and the degree of hypothetical model correlation is indicated by purple colouration. Object images reprinted with permission from Hemera Photo Objects.

about the models effects in the target region, then the RSA effects will be reduced. This reduction indicates that past responses in the source region influence the current similarity in the target region, and do so in a way specific to that model RDM being tested.

RCA revealed feedforward connectivity associated with all layers of CORnet from ~80 ms (cornet-V1, cluster p = 0.003; cornet-V2, cluster p = 0.0003; cornet-V4, cluster p = 0.003; cornet-IT, cluster p < 0.0001), with the first three layers peaking near 110-140 ms, and CORnet-IT peaking near 250 ms. Feedforward processing of semantic properties was significant later in time peaking before 300 ms (cluster p < 0.0001; Fig. 4b; Supplementary Table 4). Further, the latency of the peak feedforward effect of semantics was reliably delayed compared to all CORnet layers (V1 95% CIs [186–212 ms], V2 95% CIs [160–186 ms], V4 95% CIs [182–216 ms] and IT 95% CIs [24–60 ms]), and CORnet-IT was delayed compared to all other CORnet layers (V1 95% CIs [146–168 ms], V2 95% CIs [120–142 ms], V4 95% CIs [142–168 ms]; Fig. 4d). This shows that all types of object properties were associated with feedforward connectivity, with broad timings similar to the RSA effects presented above. Interestingly, only two types of

object properties were associated with feedback connectivity, where patterns in anterior regions influenced later patterns in posterior regions. Feedback associated with CORnet-IT was seen around 250 to 400 ms (cluster p = 0.0003), and feedback associated with semantics was seen peaking near 400 ms (cluster p = 0.0276), and again significant around 600 ms (cluster p < 0.0001; Fig. 4c; Supplementary Table 4). The timing of these feedback effects coincided with feedforward effects for the same model RDMs, suggesting a period of dynamic recurrent connectivity (both feedforward and feedback) associated with higher-level visual and semantic object properties between 250 and 500 ms. In addition, the latency of the peak semantic feedback effects (near 400 ms) was delayed compared to CORnet-IT (around 330 ms; 95% CI of peak difference [16–142 ms]; Fig. 4d). Finally, direct comparisons of feedforward and feedback peak effects showed feedforward peak effects were reliably earlier compared to feedback peaks for all model RDMs.

This shows that, at the global level of MEG sensor arrays, feedforward connectivity is associated with visual and semantic representations forming in anterior regions, in a temporal hierarchy of low, to mid, to high-level visual properties preceding

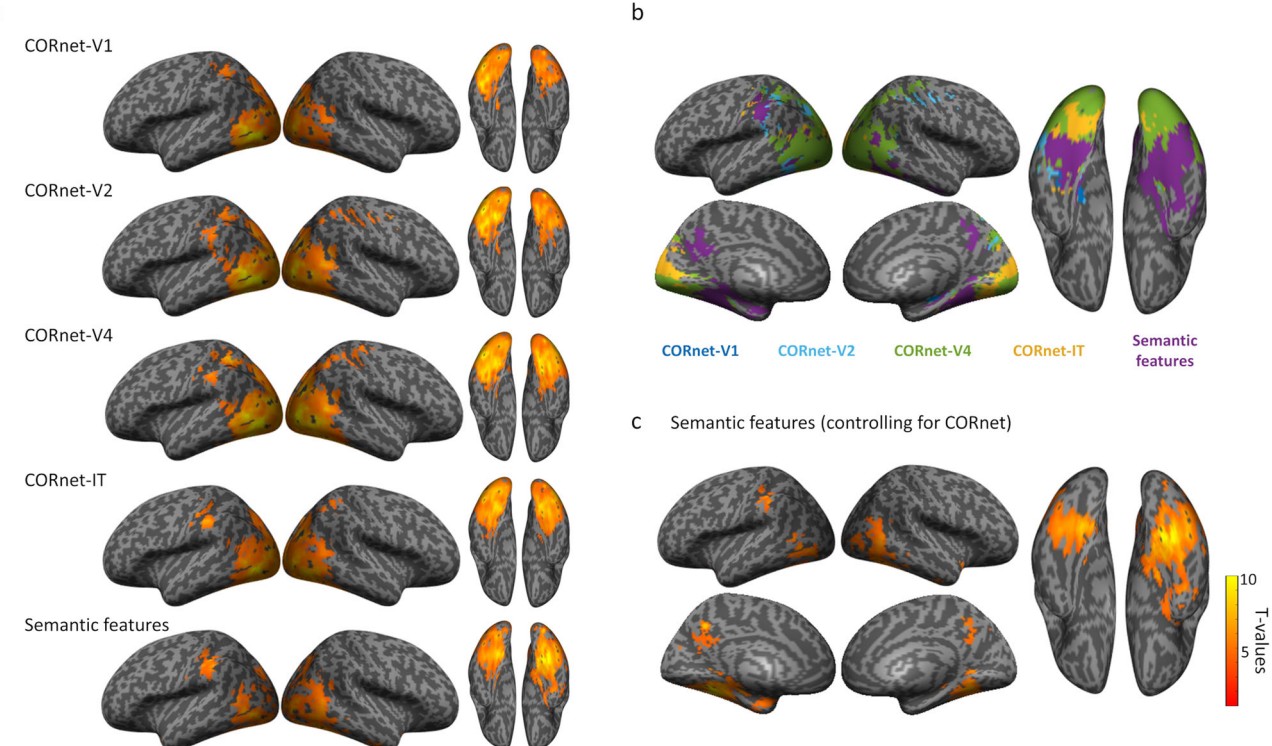

**Fig. 2 fMRI searchlight results. a** Maps show significant relationship between each model RDM and voxel patterns, voxelwise p < 0.001, cluster p < 0.05, N = 16. **b** Maps showing which model RDM had the strongest effect size at each searchlight centre voxel. **c** RSA effects of the semantic feature RDM partialling out effects of all CORnet layers.

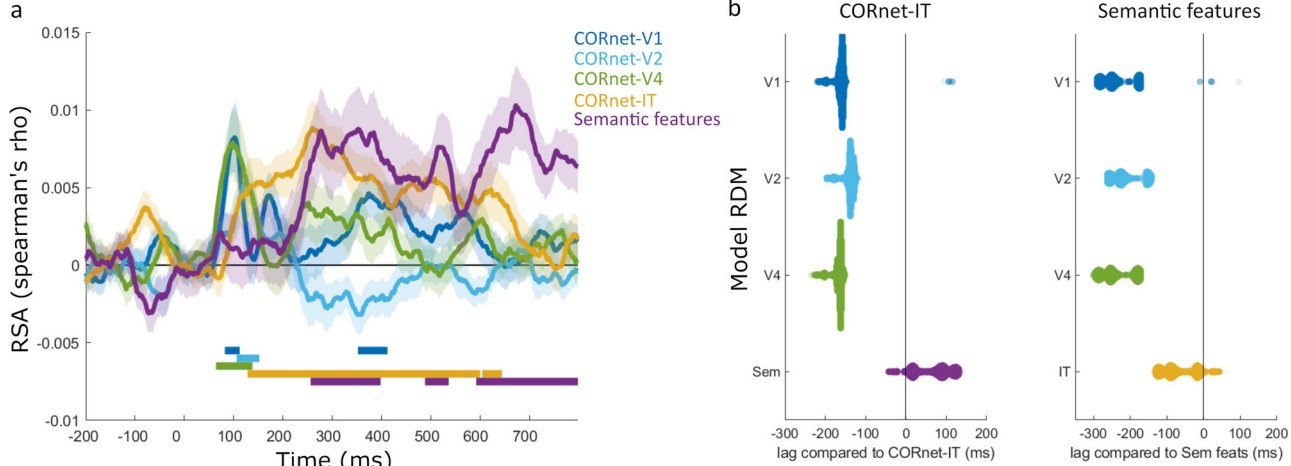

**Fig. 3 RSA results for the MEG sensor array. a** Partial correlation RSA showing the unique effects of each model RDM over time (N = 36). Shaded areas show standard error of the mean. Solid bars show time periods of significant effects. **b** Swarmplots showing the differences in peak latency between model RDMs. Distributions display resamples of the data (31,465 resamples) which were used to generate 95% CIs for the differences in peak latencies.

semantic effects. Interestingly, concomitant feedback signals were associated with higher-level visual and semantic properties pointing to the potential importance of recurrent activity during object recognition.

**MEG searchlight analysis of semantic effects (after controlling for visual models).** A core issue we address next is how semantic information is accessed through a dynamic neural system at a more detailed level than is provided by the relatively global sensor analysis. Knowing this can address what regions reflect semantic

information over time, and how connectivity aids this evolution of meaning. To this end, we conducted searchlight RSA on our MEG source-localised signals to reveal the regions and time-points sensitive to semantic structure, while partialling out any influence of the ANN CORnet layers (Fig. 5). This resulted in four significant spatio-temporal clusters that were sensitive to semantic feature information (Fig. 5a)—one covering the right posterior ventral temporal cortex, lateral temporal and extending toward the ATL, lasting between approximately 100 and 450 ms (cluster time-window = 104–442 ms, peak effect = 282 ms, p < 0.0001). A second cluster was spatially centred in left pVTC

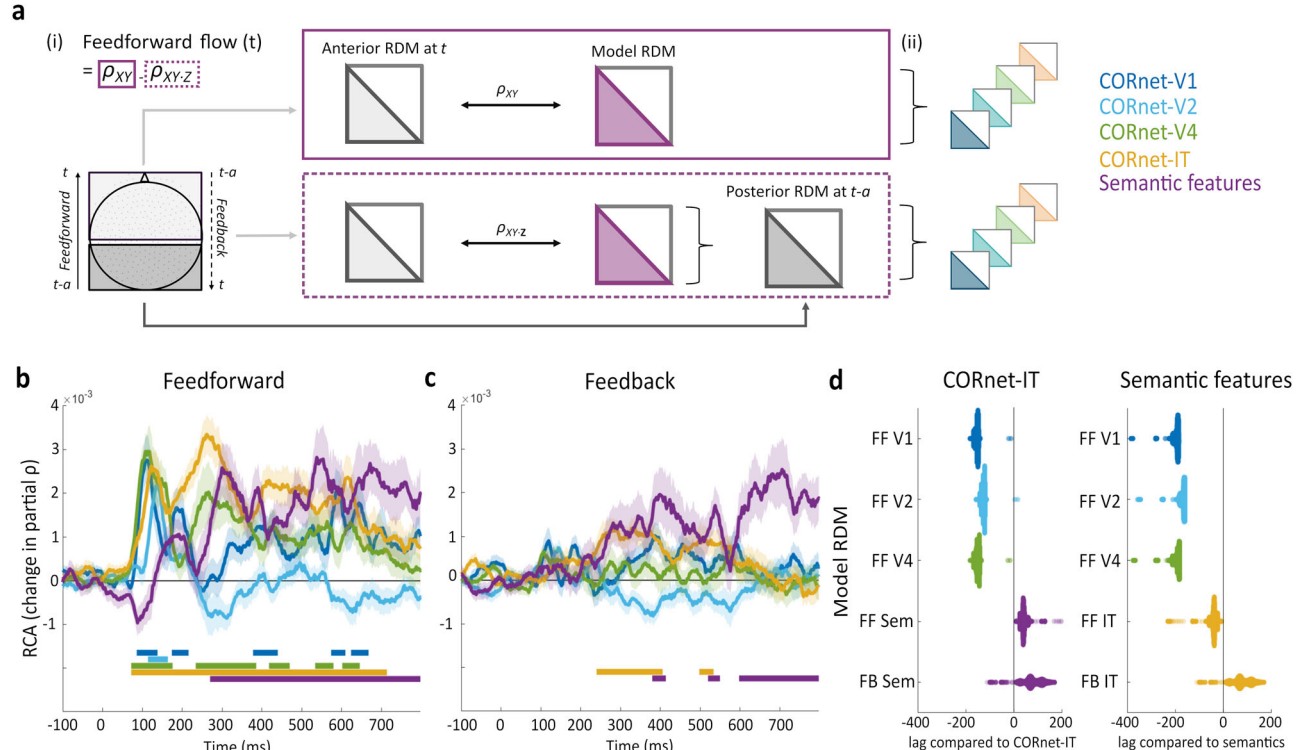

**Fig. 4 Illustration and results of the MEG sensor-level Representational connectivity analysis (RCA). a** Illustration of the calculation of feedforward information flow (feedforward connectivity at each timepoint) between anterior and posterior regions at the MEG sensor-level, as introduced by Karimi-Rouzbahani et al.[32]. i) RDMs are created from anterior and posterior sensors. Feedforward flow at each timepoint is formalised as the contribution of the earlier posterior RDM (t-30m) to the current model-anterior RDM correlation (t). This is calculated as the difference between the anterior-model RDM correlation and the anterior-model RDM correlation where the posterior RDM is partialled out. Feedback information flow is formalised as the contribution of the earlier anterior RDM (t-30) to the current model-posterior RDM correlation (t). ii) In the partial RCA, the contribution of other model RDMs is also partialled out in the calculation of both RSA timecourses. **b** Feedforward RCA effects for each model RDM. **c** Feedback effects of the model RDMs. Shaded areas show standard error of the mean. Solid bars show time periods of significant effects. **d** Swarmplots showing the differences in peak RCA latency between model RDMs. Distributions display resamples of the data (31,465 resamples) which were used to generate 95% CIs for the differences in peak latencies.

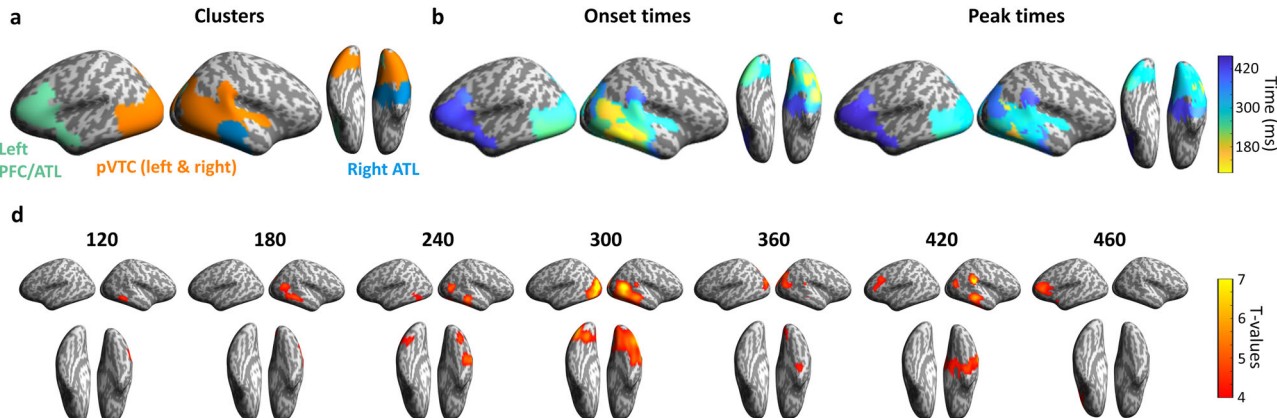

**Fig. 5 Searchlight RSA on source-localised MEG signals. a** Semantic feature effects were seen in four spatio-temporal clusters across bilateral pVTC, right ATL and left frontal/ATL. **b** Onset time of the semantic effects at each vertex and **c** time of peak RSA effect size at each vertex. **d** Maps showing significant effects of semantic features (partialling our all effects of the ANN layers) over time.

between approximately 230 and 400 ms (cluster time-window = 234–406 ms, peak effect = 306 ms, p < 0.0001). A third cluster covered the right ATL including lateral and medial aspects between approximately 350 and 450 ms (cluster time-window = 346–456 ms, peak effect = 420 ms, p = 0.008), and a fourth cluster covered left lateral ventral PFC an lateral ATL between

approximately 290 and 470 ms (cluster time-window = 386–468 ms, peak effect = 450 ms, p = 0.01).

Considering the pattern of semantic effects over space and time, we see that posterior regions appear to have earlier onsets and peak effects than more anterior regions (Fig. 5b, c), suggestive of a posterior-to-anterior progression. Significant model

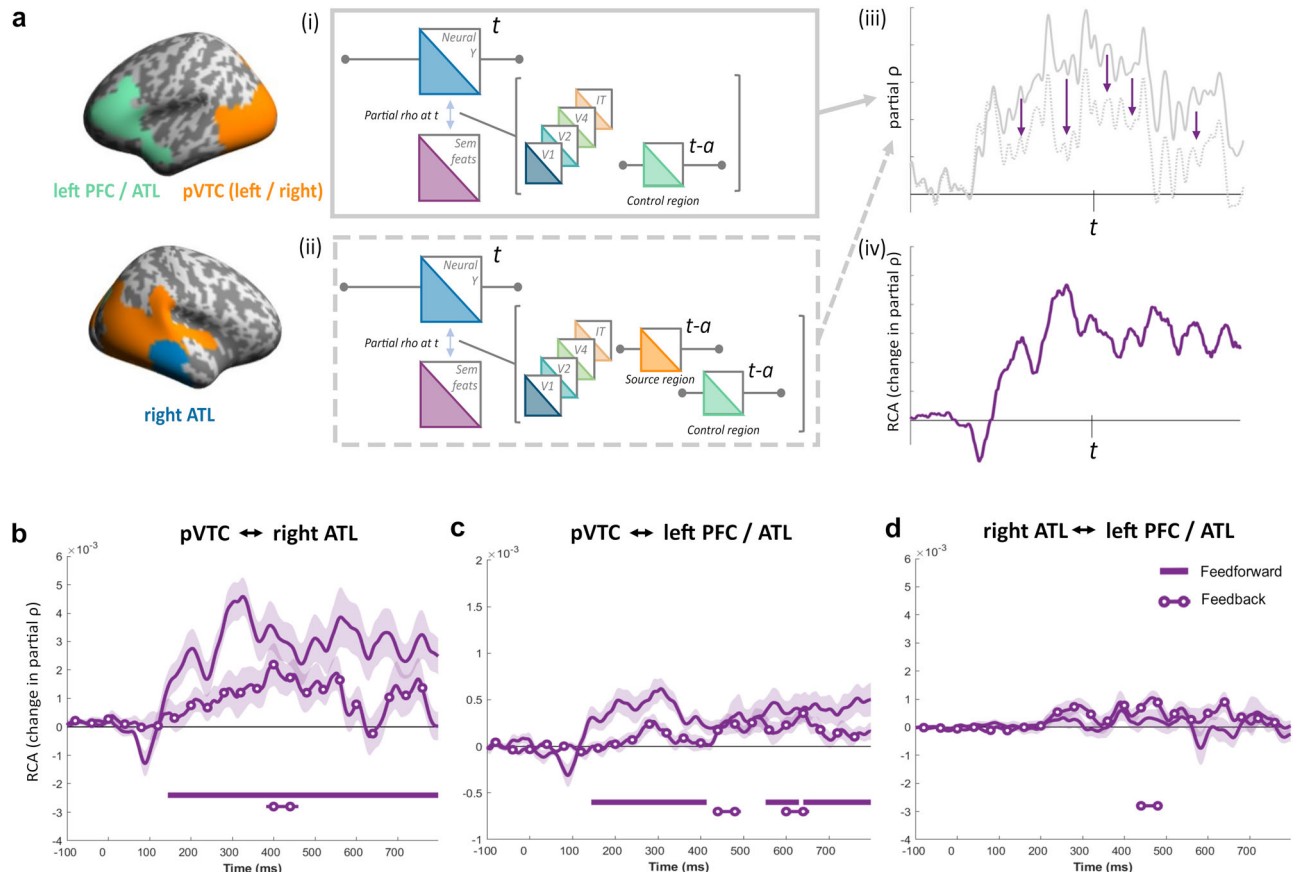

**Fig. 6 Illustration and results of RCA for source localised MEG signals. a** RCA analysis applied to ROI clusters. Feedforward effects in the source-level RCA is formalised in the same way as in the sensor-level analysis, and is based on the difference of two partial correlations. (i) The first measures the relationship between the target neural RDM and the semantic feature RDM, while controlling for other model RDMs and past RDMs from control regions. (ii) The second correlation measures the relationship between the target RDM and the semantic feature RDM, while controlling for the same other factors in addition to also removing the effects of past similarities in the source region. (iii) Example time-courses of these two partial correlations. A reduced correlation in the second partial correlation indicates the contribution of the source region to the target regions RSA effect. (iv) Subtracting the second correlation from the first is the RCA measure. **b** Effects between the pVTC and right ATL. Solid line shows feedforward effects (pvTC -> rATL) and line with circles shows feedback effects (rATL -> pVTC). **c** Feedforward (pVTC -> PFC/ATL) and Feedback (PFC/ATL-> pVTC) RCA effects between pVTC and the left PFC/ATL. **d** Feedforward (rATL -> PFC/ATL) and Feedback (PFC/ATL-> rATL) RCA effects between right ATL and left PFC/ATL. Shaded areas show standard error of the mean. Solid bars show time periods of significant effects.

correlations began in right pVTC after ~100 ms and spreads along bilateral pVTC and the right ATL within 250 ms. Semantic effects remained in the right ATL past 400 ms, alongside an additional cluster spreading across the left ventral PFC and lateral ATL (Fig. 5d). This shows semantic object processing engages a network of bilateral pVTC, ATL and the left ventral PFC within the first 600 ms of an object appearing. These effects for semantics are seen over and above those explained by the CORnet ANN of vision, with the spatial distribution of semantic effects including ventral temporal, anterior temporal and lateral occipital regions that were also present in the fMRI RSA searchlight results (see Fig. 2).

**Multivariate RCA shows feedback from anterior to posterior regions related to semantics.** We next asked how connectivity within this network related to semantic information, and visual object properties more generally. To do this we used an extended version of RCA, to show how connectivity between a pair of regions related to semantic information, while controlling for the influence of all ANN layers and the other regions of the network showing semantic effects (Fig. 6a). We created our network to consist of three ROIs based on the significant RSA clusters, being

bilateral pVTC (clusters 1 and 2), right ATL (cluster 3), and left vPFC/ATL (cluster 4). Any areas of spatial overlap between the clusters were excluded from all ROIs. Using these regions, we aimed to test the influence of past representational similarity in a source region on future RSA effects of semantics in a target region. However, we also want to control for the past representational similarity from other regions in the network, as well as CORnet. To do this we used a multivariate RCA measure which tests whether past similarity in the source region helps explain future semantic effects in the target region while also controlling for the other region in our network.

Relating specifically to semantic object properties, RCA revealed feedforward connectivity between pVTC and right ATL from ~150 ms, continuing throughout the epoch (cluster p < 0.001) as well as feedback connectivity from right ATL to pVTC near 400 ms (cluster p = 0.0058; Fig. 6b, Supplementary Table 5). We also saw feedforward connectivity between the pVTC and left PFC/ATL starting at a similar time, around 150 ms, and continuing to 400 ms (cluster p < 0.001), plus a second phase of feedforward connectivity from around 600 ms (cluster 1 p = 0.009; cluster2 p = 0.001). Small clusters of feedback from left PFC/ATL to pVTC relating to semantics were seen between 400 and 500 ms (cluster p = 0.007), and between

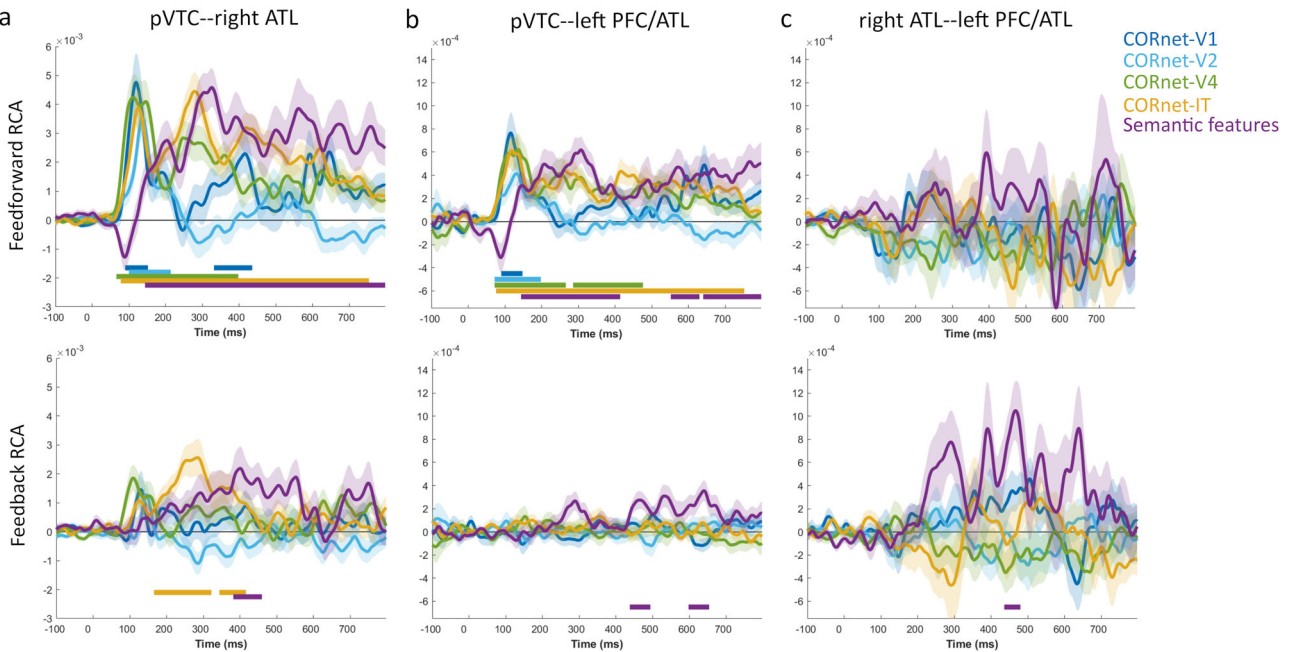

**Fig. 7 RCA effects of the visual and semantic models showing feedforward and feedback effects. a** pVTC and right ATL, **b** pVTC and left PFC/ATL, and **c** right ATL and left PFC/ATL. Shaded areas show the standard error of the mean. Solid bars show time periods of significant effects.

600 and 700 ms (cluster p = 0.009 ; Fig. 6c, Supplementary Table 5). Finally, RCA revealed no significant feedforward connectivity effects between the right ATL and left PFC/ATL region, but we did observe a small cluster of feedback from left PFC/ATL to right ATL between 400 and 500 ms (cluster p = 0.0078; Fig. 6d, Supplementary Table 5). Comparing the latencies of the peak effects, we saw that feedforward peaks occurred earlier than feedback peaks between pVTC and right ATL (Fig. 6b), except for in the case of pVTC and left PFC/ATL (Fig. 6c). While this points to prominent feedforward semantic effects from the pVTC to both ATL and frontal regions from ~150 ms, we also found evidence that feedback between all regions in the network were significant beyond around 400 ms.

Our analyses suggest that semantic processing is associated with feedforward effects from pVTC, and feedback effects to pVTC, with both the right ATL and left PFC/ATL. Direct comparisons of the RCA effect sizes between these connections revealed feedforward effects of semantics were significantly stronger for the pVTC-right ATL connection compared to the pVTC-left PFC/ATL connection between approximately 150 and 800 ms (cluster1: p = 0.005; cluster2: 249–800 ms, p < 0.001). In addition, feedback signals to pVTC were significantly stronger from right ATL compared to being from left PFC/ATL around 400 ms (cluster: 385-456 ms, p = 0.0069). While our RCA results point to feedforward and feedback effects associated with both connections, this shows that connectivity effects are stronger in the case of the pVTC and right ATL connection for semantic processing.

Finally, we extended the RCA analysis to probe how these semantic effects temporally relate to visual ANN connectivity effects. This allows us to situate the semantic effects in the context of low and higher-level visual properties. While feedforward connectivity effects between the pVTC and both anterior regions related to rapid effects of all ANN layers, the higher-level layers and the semantic model also related to later feedforward connectivity (Fig. 7, Supplementary Table 6). Most interestingly, there were limited effects of feedback relating to the visual ANN, where we only saw feedback connectivity related to the IT-layer prior to feedback relating to semantics

from the right ATL to the pVTC (Fig. 7a, Supplementary Table 6). Testing the latencies of the semantic and Cornet-IT peak effects revealed that feedback was earlier for CORnet-IT compared to semantics. This highlights that while feedback activity across this network is linked to semantic information, higher-level visual properties also relate to feedback connectivity, but at an earlier point in time compared to semantic effects. Together with the sensor analyses, our results support the notion that feedback activity to the posterior ventral temporal lobe is important for object recognition, and feedback within the temporal lobe is most relevant to the processing of higher-level visual and semantic object properties between 200 and 500 ms.

**Feedback semantic feature effects are related to behavioural response latencies.** Whilst we have shown that recurrent connectivity relates to semantic feature effects, most strongly between the pVTC and right ATL, this does not indicate whether these connectivity dynamics are behaviourally relevant. We reasoned that if this connectivity was behaviourally relevant, then changes in connectivity across participants should relate to changes in mean response times across participants. To assess this, we tested if RCA effect sizes varied according to reaction times. We used a median-split of our participants to create one group of 15 with faster mean response times, and another group of 15 with slower mean response times (overall mean naming latency 951 ms, st dev 120 ms, range 793-1169 ms). This allowed us to determine if feedforward or feedback RCA effects significantly varied depending on behaviour.

Contrasting RCA effects between these two groups of participants revealed that participants with slower responses displayed significantly greater feedback RCA than those with faster responses. This was observed between the right ATL and the pVTC for the semantic model RDM (time window = 384-420 ms, cluster p = 0.0386; Fig. 8). These effects overlap in time with the feedback RCA effects of semantics seen for the whole group. There were no further behavioural effects seen for feedforward or feedback RCA. Additional exploration of the

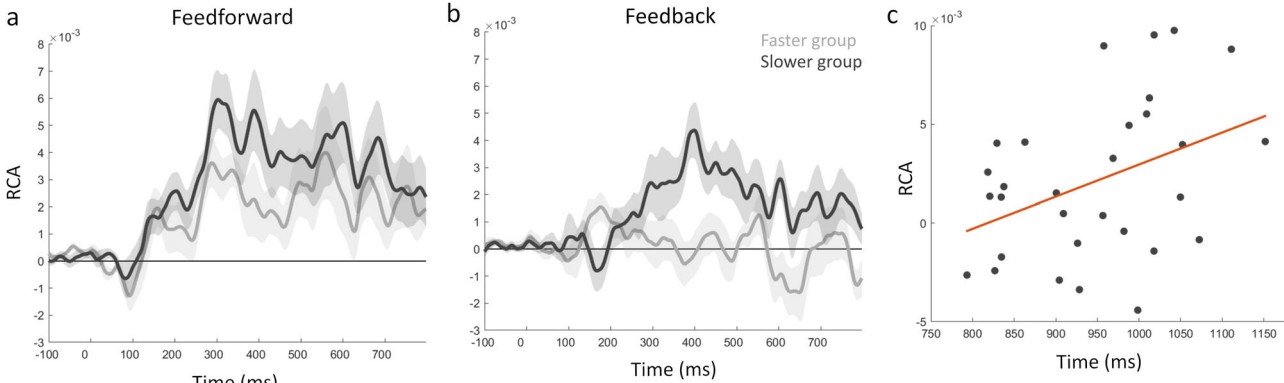

**Fig. 8 Relationship of RCA effects to behaviour. a** Feedforward RCA effects of the semantic model for the faster group (light grey) and slower group (dark grey). **b** Feedback RCA effects for the two groups, where feedback is significantly increased for the slower group. Shaded areas show the standard error of the mean. **c** Correlation plot showing the relationship between response time and Feedback RCA for each participant, and the fitted linear effect.

CORnet model layers also revealed no significant modulations of RCA by behaviour.

## Discussion

Visual object recognition is a dynamic and interactive process. However, our lack of understanding of the functional role that the interplay between feedforward and feedback connectivity plays, and the unique contributions visual and semantic object properties make to this means our cognitive and computational models of visual object recognition have remained limited. Artificial neural networks that include across layer feedback are claimed to provide a better account of object processing compared to feedforward neural network models[31,68,69], while brain activity within the ventral visual pathway has been shown to display long-range recurrent dynamics[12,25,29,70–72]. But despite highlighting that recurrent activity might play an important role in models and the brain, a critical question for understanding the neural mechanism of object recognition is what the functional relevance of recurrent activity in the brain is. Here, our research makes two important advances to enable us to address this vital question. First, using RSA and RCA combined with an ANN model of visual perception and a semantic feature model, we showed that different visual and semantic properties of objects related to time-resolved feedforward and feedback signals in the VVP and frontal lobe. Importantly, we linked recurrent connectivity to specific cognitive measures of higher-level perceptual and semantic features and showed that this measure of recurrent connectivity relates to behaviour. Second, we demonstrated the importance of semantic object information in explaining evoked neural activity and connectivity in the VVP and frontal lobe, illustrating the contributions made by semantic features beyond that explained by ANN models. In this manner, we suggest that recurrent activity might support object recognition in the brain, specifically relating to higher-level visual and semantic properties between ~200 and 500 ms, driven predominantly by interactions between the anterior and posterior ventral temporal cortex.

Many traditional models of visual object recognition have stressed that 'core' aspects of object recognition are supported through feedforward processes along the VVP[2,19]. More recently, long-range feedback and recurrent activity has been acknowledged as being important for object recognition[1,18,23,24,26,28,29,31,73], although the situations when recurrent activity is more prevalent, and what its functional cognitive role is, have been less clear. Our results make an important novel contribution to this issue – namely in specifying the role of long-range, between-region recurrent activity. Using

RSA, RCA and quantifiable models of visual and semantic object properties, we were able to track the spatio-temporal evolution of object properties, and how different properties related to feedforward and feedback connectivity both within the VVP, as well as between temporal and frontal regions.

We showed that higher-level visual properties of the ANN (CORnet layer IT) and semantic feature measures are both related to feedback connectivity from the ATL to the pVTC, with higher-level visual feedback preceding semantic effects. These feedback effects overlapped in time with feedforward connectivity effects of the same object properties, suggesting a period of dynamic interactions between the pVTC and the ATL in support of higher-level visual and semantic representations. In addition, we saw more transient effects of feedback from left PFC/ATL to right ATL, and left PFC/ATL to the pVTC beyond 400 ms. This suggests that recurrent activity in the VVP, with support from the PFC, enables higher-level visual and semantic object representations, and does so even for highly familiar, unambiguous objects like those used here.

Our connectivity analysis established a distinct later period of feedback captured uniquely by the ANN layer-IT and the semantic feature model, predominantly beyond around 200 ms. This is in line with the view that recurrent processing is crucial for the transition from lower-level visual to higher-level visual and object-specific semantic representations[12,25,27,74–76]. According to this view, variations of which were posited by multiple-state interactive (MUSI) theory[73,77] or the top-down-facilitation model[24,70,78], an initial feedforward sweep activates visual and coarse categorical representations, before an interplay of higher-level feedforward and feedback signals serves to build object-specific semantic representations at 200–500 ms[79]. Previous connectivity studies have shown that feedback signals may become increasingly active between 200 and 400 ms[70] and are modulated by semantic task demands[25,76]. Further, semantic properties have also been related to feedforward, feedback, and lateral interactions within the VVP[12]. Here, we revealed that high-level visual and semantic information are uniquely related to these dynamic recurrent interactions and that feedback signals relating to semantic effects were modulated with behaviour. Furthermore, whilst being staggered in time, feedforward and feedback flows overlapped, which may be in line with stochastic evidence accumulator models of perceptual decision-making during object recognition, with decisions made on continually evolving representations through recurrence, and recognition triggered when a threshold has been reached[80–82] as indicated by the relationship between feedback and behaviour.

Our research helps expand our understanding of the circumstances under which recurrent activity occurs during object recognition. It had been suggested that recurrent activity in the VVP increases when the task requires a more semantically detailed response[25], while other research suggests that top-down effects within the VVP, or from frontal regions, increases under challenging object recognition conditions such as occlusion and degraded stimuli[18,20,22,23,77]. Given the lack of challenging conditions here, this suggests that while recurrent activity supports higher-level visual and sematic processing even in non-challenging situations, processing of these object properties, and indeed lower-level ones, may be facilitated to a greater extent by feedback activity when the situation dictates. We also saw that participants who were slower at responding, showed comparably increased feedback activity, which might suggest that such tasks can be performed with minimal feedback if speeded responses are required[83]. Further studies will be needed to more directly assess how the degree of feedback varies across individual concepts, image manipulations and tasks. For instance, we would predict that more semantically confusable items would require more feedback between the ATL and pVTC.

Our searchlight RSA analysis of source-localised MEG signals pointed to a dynamic network underpinning semantic representations involving bilateral pVTC, the ATL and left frontal lobe. In addition to feedback connectivity linked to semantics between the ATL and pVTC, significant periods of feedback connectivity beyond 400 ms from the frontal lobe were also linked to semantic object properties. Between 400 and 500 ms, feedback connectivity from the left frontal lobe was seen to both the right ATL and pVTC, with an additional feedback cluster from left frontal to pVTC beyond 600 ms. We also saw evidence of a feedforward effect of semantics from the pVTC to both the ATL and frontal lobes beginning before 200 ms, suggesting that semantic representations are supported by early feedforward activity from pVTC to the ATL and frontal lobe, and later feedback activity from the frontal and ATL, to the pVTC, with an additional period of feedback from the frontal to the ATL.

While semantic responses to visual objects are a dissociable form of information compared to image-based visual object properties, there have been some suggestions that semantic effects seen in neuroimaging might be partially explained by visual properties of the stimuli. While a clear distinction between higher-level visual and semantic object properties can be problematic and is a potentially futile endeavour in some regards (e.g. is the knowledge that something has stripes purely distinct from cells tuned to detect striped textures), there are certainly semantic properties which transcend vision. Here, we show that visual object properties, captured through an ANN, and semantic properties, display different spatial, temporal and morphological RSA and RCA effects across both the MEG and fMRI datasets. Both our fMRI RSA searchlight and the MEG-based searchlight showed spatially similar patterns where semantic properties related to activity patterns in bilateral pVTC and the ATL beyond that which the visual ANN could explain. Our MEG results further showed that semantic effects tended to lag behind higher-level visual effects, as expected considering they reflect a more abstracted type of object information. This relationship between visual and semantic effects is similar to our previously reported work[11,12] and a recent study by Jozwik et al.[16]. While there is broad consistency in the timing of peak effects for the ANNs, there is more variability comparing our semantic feature effects to their visuo-semantic model. The MEG RSA results Jozwik et al.[16], present show earlier peak effects for a visuo-semantic model, approximately 150-250 ms, compared to our peak semantic effects between 200 and 400 ms, in line with a large body of work on N400 semantic effects (see[84]). The visuo-semantic model Jozwik used includes various visual features of colour, texture along with category labels (e.g. giraffe, food, kiwi) for

the object images which may create an earlier shift in time. The semantic model we use is a relational model, which does not include any category labels or taxonomic information as these are not usually considered true semantic features in models of conceptual representations. Our semantic feature model, based on over 3000 features given to concepts presented as words, captures the similarities and differences between a broad set of concepts and is likely more abstracted from the visual images, contributing to the delayed model correlations. A final factor additionally contributing to the earlier visuo-semantic effects is the large number of stimuli repetitions used by Jozwik, compared to our single presentation. Semantic effects are known to change in amplitude and latency upon repetition (e.g.[85]) although this is likely to influence visual effects less so.

We would suggest that a visual-to-semantic transition, in terms of its representational state and how it relates to perceptual and cognitive properties, occurs gradually within the first 400 ms, with previous MEG connectivity evidence suggesting recurrent connectivity supports this translation, including feedback from the ATL to the pVTC[12]. Together with previous connectivity, fMRI, and neuropsychological studies, our results underpin the role of feedback from the ATL to the pVTC in supporting the semantic aspects of visual object recognition[12,74,75,86], with additional feedback contributions from the frontal lobe[24,76,87–89]. While current accounts might point to differential effects of feedback either within the temporal lobe, or between frontal and temporal regions[73,89–91], the approaches we outline here could be used to further understand such relative contributions to semantics and behaviour in finer detail.

Finally, consistent with many accounts of 'core' object recognition[2,19,92], earlier layers of the visual neural network model related to rapid feedforward connectivity effects between posterior VTC and both the ATL and frontal lobe. In relation to higher-level visual and semantic effects, this points to feedforward connectivity supporting a representational shift through low to high-level visual features and semantics, prior to recurrent effects of higher-level visual and semantic object properties. The current work has implications for future developments of brain inspired ANNs. Our work aligns with computational work suggesting ANNs are improved by between-layer feedback connections in addition to the within-area recurrent (or lateral) and skip connections[31]. For models such as CORnet, which makes links between network areas and neural regions, our work suggests that they would more accurately account for object recognition through the addition of an 'ATL' layer with nodes trained to reflect conceptual structure (as in[42]), and with feedback from the ATL to other layers[68]. Such extensions could also track the stimulus history to enable models to account for priming effects and MTL-like behaviours[93].

Visual object recognition must be supported by dynamic and interactive neural mechanisms, which require techniques suited to tracking neural activity and connectivity at a fine-temporal resolution, combined with cognitive models of our stimuli. Here, we show that long-range recurrent activity and connectivity underpins object recognition in the VVP, and outside. Through specifying models of both the visual and semantic aspects of our stimuli, we saw that recurrent activity within the VVP related to higher-level visual properties prior to semantic object properties, with additional feedback connectivity from the frontal lobe relating to semantics. These results highlight the important contributions of both visual and semantic object properties to the evolving object representations, and that visual models alone cannot account for all aspects of visual object recognition.

## Methods

Here we combined participants from two MEG datasets that were previously reported by Clarke et al.[11,12], and Bruffaerts et al.[94],

both using the same stimuli, task, and acquisition parameters. We also report a re-analysis of a previously published fMRI dataset[48]. Therefore, only the main aspects of data acquisition and pre-processing are reported here.

**MEG participants and procedure**. The study used a total of 36 participants, 15 from Clarke et al.[11,12] and 21 from the young group in Bruffaerts et al.[94]. Data from Bruffaerts et al. was part of the Cambridge Centre for Ageing and Neuroscience (CamCAN) study, and the data used in the preparation of this work can be obtained from the CamCAN repository (available from https://www.cam-can.org)[95,96]. Informed consent was obtained from all participants and ethical approval for the studies were obtained from the Cambridgeshire Research Ethics Committee. All experiments were performed in accordance with relevant guidelines and regulations. All ethical regulations relevant to human research participants were followed.

All participants performed a basic-level naming task during which they were asked to identify 302 common objects from 11 superordinate categories, comprising of both living and non-living things. All objects were shown to participants in a pseudo-randomised order and presented in colour on a white background. A trial began with a fixation cross for 500 ms, before the object was presented for 500 ms, after which followed a blank screen for a random interval between 2400 ms and 2700 ms. Each object was only shown once. Incorrectly named trials, where participants could not name or incorrectly named the object, were excluded from all further analyses (mean number of remaining trials was 270).

**MEG recording/MRI acquisition**. All MEG data were collected at the MRC Cognition and Brain Sciences Unit, Cambridge, UK, using a whole-head 306 channel (102 magnetometers, 204 planar gradiometers) Vector-view system (Elekta Neuromag, Helsinki, Finland). Blinks and eye movements were recorded using electro-oculogram (EOG) electrodes and the head position was recorded using five Head-Position Indicator (HPI) coils. Participants' head shapes, and the positions of EOG electrodes, HPI coils, and fiducial points (nasion, left and right periauricular) were digitally recorded with a 3D digitiser (Fastrak Polhemus, Inc., Colchester, VA, USA). MEG signals were recorded at a sampling rate of 1000 Hz and a high-pass filter of 0.03 Hz. To enhance source localization, T1-weighted MP-RAGE scans with 1 mm isotropic resolution were acquired for each subject using Siemens 3-T Tim Trio.

**MEG pre-processing**. Initial pre-processing of the raw MEG data used MaxFilter (version 2.2) to apply temporal signal space separation (tSSS) and head-motion correction. The MEG signals were then low-pass filtered at 200 Hz using a fifth-order Butterworth filter, and high-pass filtered at 0.1 Hz using a fourth-order Butterworth filter in SPM12 (Wellcome Institute of Imaging Neuroscience, London, UK). The data were then epoched between -1000ms to 1000 ms and down-sampled to 500 Hz. Artefact removal was performed using Independent Component Analysis (ICA) implemented with RUNICA in EEGlab[97], supplemented with SASICA[98] to identify muscle and speech-related artefacts. Finally, baseline correction was applied using the pre-stimulus time window of -200 to 0 ms. Source localisation of MEG signals used a minimum-norm procedure based on both magnetometer and gradiometer sensors. Individual participants MRI images were segmented and spatially normalised to an MNI template brain consisting of 5124 vertices, which was inverse normalised to the individuals specific MRI space. MEG sensors were co-registered to the MRI image using the three fiducial

points and the additional headpoints. A single shell forward model was used.

**MEG representational similarity analysis (RSA)**. Representational Similarity Analysis[35] was used to test model representational dissimilarity matrices (RDMs) capturing visual and semantic information against neural activity patterns as measured by MEG, at both the sensor-level and for a spatiotemporal searchlight analysis[99].

**Construction of model RDMs**

*Visual RDMs*. Visual model RDMs were created using CORnet-S, a four-layer recurrent DNN for core object recognition which was modelled on the anatomy of the primate ventral visual stream, encompassing areas V1, V2, V4, and IT[60] and trained on ImageNet. CORnet-s has been shown to perform well in predicting both neural responses recorded in macaque V4 and IT, as well as human behavioural similarity judgements[28,60]. The nodal activations for network layers V1, V2, V4, and IT were extracted for each object using the THINGSVision toolbox in Python[62]. After extracting the activations for each of the 302 objects, an RDM was created for each layer by calculating the Pearson's correlation distance between all possible pairs of pictures resulting in a 302 × 302 symmetrical dissimilarity matrix per layer. Here we used a single pre-trained instance of CORnet-S, however, it has been suggested that nodal activations vary depending on the initial random seed of the network[100]. While it is unclear if this variability would influence our RSA results (or those of other studies), we show the generalisability of our MEG semantic effects against a selection of different ANN architectures (see Supplementary Fig. 2). Specifically, we also created model RDMs for CORnet-RT (areas V1, V2, V4 and IT), AlexNet, Resnet50 and VGG19, where activations were extracted using THINGSVision[62].

*Semantic RDM*. The degree of semantic similarity between objects was quantified based on semantic features, using the Centre for Speech, Language and the Brain property norms[63], containing 836 different concepts and their association to 3026 different properties (excluding taxonomic features, which refer to superordinate categories and are not typically regarded as semantic features in studies of conceptual representation). Using these property norms, each object could be represented by a list of features that collectively define its conceptual content. A 3026-long vector was created for each object, consisting of zeros and ones indicating whether each specific feature was associated with that object. An RDM was then created by calculating the Cosine distance between all possible object pairs resulting in a symmetric 302 × 302 matrix.

**Statistics and reproducibility**

*MEG sensor RSA*. RDMs were created in order to calculate the similarity between MEG sensor activity patterns associated with each object, for each participant and timepoint. RDMs were derived from spatiotemporal MEG sensor patterns extracted over a 40 ms sliding window centred around each timepoint, $t$, recorded from 204 planar gradiometers. 1-Pearson correlations were calculated pairwise for all combinations of object pairs, resulting in a 302 × 302 dissimilarity matrix for each timepoint.

In order to assess if the MEG RDMs related to visual or semantic object properties, the MEG RDMs were correlated with each model RDM at every timepoint (using only the vectorised upper triangle), resulting in an RSA time-course for every model RDM and for each participant. Partial Spearman's correlations were used to test between a single model RDM and the MEG signals, controlling for the influence of all other model RDMs in order to find the unique effects of each model RDM.

The group-level RSA was assessed for statistical significance using one-sample t-tests against a baseline of zero ($\alpha = 0.01$) for each timepoint and each model RDM. Cluster-mass permutation tests were applied in order to control for multiple comparisons across the number of time-points tested[101]. Each cluster of timepoints for which the correlation between MEG RDM and model RDM was significant was assigned a cluster p-value based on the sum of the t-values within the cluster. To determine these cluster p-values, a null distribution was created from the permuted RSA time-courses, where the sign of each subject's RSA time-course was randomly flipped before conducting the one-sample t-test again and retaining the cluster size of the largest cluster, which was added to the null distribution. This was repeated for 10,000 random permutations of the data. The cluster p-value for each of the original clusters was defined as the proportion of the 10,000 permutations (plus the observed cluster) that were greater than or equal to the observed cluster-mass.

**Peak latency analysis**. In order to assess the statistical significance of the difference in peak latencies, we used a leave-n-out jackknife approach, with n set to 4 (12% of the data; highly similar results are seen at $n = 3$ and $n = 5$). The leave-n-out approach was used in pace of bootstrapping to create a large number of resamples of the data from which confidence intervals for the differences in peak latencies can be calculated. For each of the possible 31,465 unique resamples of the data, group average RSA time-courses were calculated and the timepoint of the maximum effect was extracted. This created a distribution of peak times for each model RDM. The distribution of peak was then used to define 95% confidence intervals (CIs) of the peak, and distribution of peak differences to define 95% CIs of pairwise differences. The null hypothesis was rejected if the 95% CIs of the pairwise differences did not include zero[102].

**Searchlight RSA**. The relationship between the representational geometries of the cognitive RDMs and of the MEG signals was tested using spatiotemporal searchlight RSA[99] of the source localised MEG signals. At each vertex and time-point, $t$, single-trial MEG signals were extracted for all vertices within a 10 mm radius for timepoints within a 40 ms time window centred at $t$. RDMs were calculated as 1-Pearson's correlation between all pairs of single trial responses. This procedure was repeated for all vertices and timepoints between -200 and 500 ms (at 2 ms increments). Here we focus on the semantic model, therefore the MEG data RDMs for each point in space and time were compared against the dissimilarity values computed for the semantic model RDM using Partial Spearman's correlation, controlling for the CORnet-S layer model RDMs (using the upper triangles only), resulting in an RSA timeseries for each vertex, timepoint, and participant.

Random effects analysis testing for positive RSA effects was conducted for each timepoint and vertex using one-sample t-tests against zero (alpha 0.01). To correct for multiple statistical comparisons over time and space, a second level of statistical thresholding was applied using cluster-based permutation testing after clusters were formed by contiguous above threshold effects in space (defined by adjacent vertices on the cortical mesh) and time. The size of the observed clusters was calculated as the sum of the t-values for time/spatial points within the cluster. P-values were assigned to each cluster using a null distribution based on permuted MEG signals. For each of the 1000 permutations, the sign of the correlations was randomly flipped for each participant before one-sample t-tests of the permuted data at each vertex and timepoint, and the largest cluster (sum of above-threshold t-values) over time and space was retained for the null distribution. The cluster p-value for each of the observed clusters was defined as the

proportion of the 1000 permutations (plus the observed cluster-mass) that were greater than or equal to the observed cluster-mass.

**Representational connectivity analysis (RCA)**. To assess whether information captured by our models of visual and semantic processing is transferred between regions during object recognition, we applied an RSA-based informational connectivity analysis developed by Karimi-Rouzbahani et al.[32]. Representational connectivity analysis (RCA) works by establishing the contribution of earlier (in time) neural representations from the source region to current informational content in the target region, thereby uniquely providing information about the latency, direction, as well as the representational content of information flow. This analysis was applied to both the sensor and source-localised MEG signals.

**Sensor RCA**. To approximate anterior and posterior brain regions, MEG gradiometer sensors were split into two equal sensor regions according to the sensor y-coordinate. MEG RDMs were then calculated for each region separately, at each timepoint and for each subject. However, in contrast to the RSA analysis, MEG RDMs were based on a single timepoint rather than a 40 ms time-window to make sure time-lagged posterior and anterior RDMs did not share any of the same data, and the upper triangle of each RDM was converted into a representational dissimilarity vector (RDV). RCA between a source and target region was calculated for each timepoint, $t$, and each model RDV in two steps. First, we calculated the partial Spearman's correlation between the target MEG RDV and a single model RDV controlling for all other model RDVs. Second, we calculated a second partial Spearman's correlation between the target MEG RDV and a single model RDV, controlling for all other model RDVs (as in the first calculation), but additionally controlling for the influence of the source MEG RDVs from previous timepoints. The time range of the previous RDVs was set to include -30 ms to -2 ms, chosen based on previous studies investigating the timeframe of activation flow between sensory and frontal regions[32,103]. RCA was defined as the difference between the first and second calculation, with feedforward RCA being when the posterior region was the source and the anterior region was the target, and feedback RCA being when the anterior region was the source and posterior region was the target.

**Source-level RCA**. RCA was also applied to source-level regions of interest. Regions were derived from the spatiotemporal clusters revealed from the searchlight RSA analysis. Each region could act as a source and target to the other regions, resulting in three feedforward and three feedback RCA time-courses for each model RDM and participant. In order to ensure bivariate effects could not be explained by a common effect from the third region, the third region was added to the partial correlation analyses. For example, to ensure the RCA effects between Region A and Region B could not be explained by past effects in Region C, the Region C MEG RDM was partialled out in both the first and second partial Spearman's correlations.

For both the senor and source-level RCA analyses, feedforward, and feedback RCA time-courses for each model RDM and each participant were created, which were assessed for statistical significance using the same procedures outlined above for the senor RSA analysis and peak latency analyses.

*fMRI participants and procedure*. We re-analysed fMRI data, originally reported in Clarke and Tyler[48], collected from 16 participants (10 female, 6 male). All participants gave informed consent and the study was approved by the Cambridge Research Ethics Committee. All ethical regulations relevant to human research participants were followed. The data pre-processing and

RSA analysis framework were the same as in Clarke and Tyler[48] and so only the main aspects of the experiment are reported here.

Participants named 145 objects, where each object was depicted by a photograph of an isolated object on a white background. From the 145 objects, 131 were from one of six object categories (animals, fruit, vegetables, tools, vehicles, musical instruments), and the remaining 14 objects did not clearly belong to a category and were not included in the analyses. Each trial consisted of a fixation cross lasting 500 ms, before an object was presented for 500 ms followed by a blank screen lasting randomly between 3 and 11 s. All objects were presented once in each of the six fMRI scanning blocks.

*fMRI scanning and preprocessing.* Participants were scanned at the MRC Cognition and Brain Sciences Unit, Cambridge, in a Siemens 3-T Tim Trio MRI scanner (Siemens Medical Solutions, Camberley, UK). There were 3 functional scanning sessions using standard gradient-echo echoplanar imaging (EPI) sequences with $3 \times 3 \times 3$ mm voxel size. Prior to functional scanning, a high-resolution structural MRI image was collected using an MPRAGE sequence with 1 mm isotropic resolution. Preprocessing of the functional data consisted of slice-time correction and the spatial realignment of the functional images only using SPM8 (Wellcome Institute of Cognitive Neurology, London, UK). The un-normalised and non-smoothed EPI images were entered into a general linear model to obtain a single t-statistic image for each concept picture based on all 6 repetitions. Activity in these t-maps was then restricted to voxels identified as grey matter, as identified from the T1 images, and used in subsequent representational similarity analysis.

*fMRI searchlight RSA.* An RSA searchlight mapping procedure[104] was implemented using the RSA toolbox[59] in Matlab to determine if object dissimilarity predicted by the visual and semantic models was significantly related to dissimilarity defined by local fMRI activity patterns. At each voxel, object activation values from grey matter voxels within a spherical searchlight (radius 7 mm) were extracted to calculate distances between all objects (using 1 - Pearson correlation) creating an object dissimilarity matrix based on that searchlight. This fMRI RDM is then compared to each model RDM (using Spearman's rank correlation) and mapped back to the voxel at the centre of the searchlight.

For group random-effects analyses, the Spearman's correlation maps for each participant were Fisher-transformed, normalized to standard MNI space, and spatially smoothed with a 6 mm FWHM Gaussian kernel. Maps were entered into a random effects analysis (RFX) in SPM12 where each voxel was tested using a one-sampled t-test against zero. Voxelwise multiple comparisons correction was applied through a voxelwise threshold of $p < 0.001$ and FWE-cluster $p < 0.05$.

**Reporting summary**. Further information on research design is available in the Nature Portfolio Reporting Summary linked to this article.

## Data availability

The data used in this research was obtained from different experiments with different availabilities. The MEG data collected as part of the Cambridge Centre for Ageing and Neuroscience was part of stage III and is available upon requested (see https://camcan-archive.mrc-cbu.cam.ac.uk/dataaccess/). The remaining MEG data is found here https://osf.io/2uqf4/[105], and fMRI data here https://osf.io/e2s59/[106]. The source data to produce the figures is hosted at the same OSF site.

## Code availability

The custom code used for the MEG RSA analysis is available here - https://github.com/AlexDClarke/MEG_RSA_VonSeth.

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

## Acknowledgements

This research was funded in whole, or in part, by the Wellcome Trust [Grant number 211200/Z/18/Z to A.C.], and the European Research Council under the European Community's Seventh Framework Programme (FP7/2007-2013)/ERC Grant agreement no. 249640 to L.K.T. The Cambridge Centre for Ageing and Neuroscience (Cam-CAN) research was supported by the Biotechnology and Biological Sciences Research Council (grant number BB/H008217/1). We thank the Cam-CAN respondents and their primary care teams in Cambridge for their participation in this study. We also thank colleagues at the MRC Cognition and Brain Sciences Unit MEG and MRI facilities for their assistance. For the purpose of open access, the author has applied a CC BY public copyright licence to any Author Accepted Manuscript version arising from this submission.

## Author contributions

L.K.T. and A.C. designed the study, L.K.T. and A.C. collected the data, A.C. supervised the project, J.v.S., V.I.N., and A.C. analysed the data and J.v.S., L.K.T., and A.C. wrote the manuscript.

## Competing interests

The authors declare no competing interests.
