## [Peer Review File · Communications Biology]

Reviewers' comments:

Reviewer #1 (Remarks to the Author):

1. In general, how do you explain the ~150ms lag between the peak time of the first three layers of the CORnet and the CORnet-IT? It seems pretty long, in macaque, usually IT and V4 has about 50ms latency.
2. In general, semantic features has peak correlation with the brain data after ~300ms. Humans are able to recognize objects that are briefly presented (~50ms), meaning it does not always take 300ms to "access semantic memory." Is this the nature of MEG data?
3. The study is very similar to a study published from the Kriegeskorte group (<https://www.jneurosci.org/content/jneuro/43/10/1731.full.pdf>; Jozwik et al.) except that the current study tries to disambiguate between the contribution of the feedforward and feedback processing. The paper that I cite here also compares CORnet-RT to brain data. It seems natural to try this model as well. There are some differences in the peak latency between the studies. How do you explain the discrepancy? Also, in order to claim that semantic features explains beyond the vanilla object recognition model, it would be better to test whether including the semantic model actually explains more of the variance in the brain data. Jozwik et al. does this by fitting the brain data to different GLM models (one with both CORnet and semantic model and the other with just the CORnet).
4. Are there images that elicited higher feedback signals more than other images? If so, are there any image properties that predict the engagement of feedback?
5. Are there any differences in brain signals for the incorrectly labeled trials? Either these mistakes are from perceptual or semantic errors. It would be interesting to see if you can see this in the brain data. For example, if these were perceptual errors (subjects didn't see them, or not paying attention could look at eye movement patterns) then the perceptual areas in the brain could have been less active. Also, you can compute if either the feedforward or the feedback processing is affected.

Reviewer #2 (Remarks to the Author):

In the paper "Recurrent connectivity supports higher-level visual and semantic object representations in the brain", von Seth et al. use neuroimaging and computational modelling to investigate the spatiotemporal dynamics during object recognition.

The authors differentiate between semantic and visual processing (processing the visual attributes of the image vs. semantic object processing). They investigate how the brain's representation of an object changes over time, and how that relates to both the visual and semantic aspects of an object. This paper presents an ambitious attempt to elucidate the neural mechanisms underlying visual object recognition, emphasizing the interplay between visual and semantic properties and the role of feedforward and feedback signals. The methodological approach is commendable, using techniques like Representational Similarity Analysis (RSA), Recurrent Connectivity Analysis (RCA), and implementing Artificial Neural Networks (ANN).

The topic is timely and of interest to the community. However, I believe the study's core contributions and aims are obfuscated due to the varying angles of introduction and unclear research question, which limits the overall impact of the paper. Below I highlight a few concerns that, I hope, will help the authors further strengthen their submission.

Specific Comments:

1. I found it difficult to understand what the aim or research question of the study was, as the authors introduce the topic from different angles. This makes it challenging to evaluate the impact of the

current findings.

I believe the authors have made interesting contributions, in showing for example that recurrent activity was linked to higher-level visual and semantic properties and was mainly driven by interactions between the anterior and posterior ventral temporal cortex. The research also highlighted the importance of semantic object information in explaining neural activity, thereby proving that semantic contributions go beyond those explained by ANNs alone (and it suggests a more dynamic and interactive model of semantic activation than perhaps previously proposed). However, in the current form I believe the core findings and contributions are obfuscated and it is for the reader not immediately clear what to do with this information or how this goes beyond what we already know from previous literature.

2. Related, I found the paper to be missing references to and a discussion of previous experimental and computational work from various labs that investigate the functional role of recurrent computations during object recognition (or the incorporation of semantic features). To list a few: <https://doi.org/10.1002/brb3.1373>, <https://doi.org/10.1523/JNEUROSCI.1424-22.2022>, <https://doi.org/10.1371/journal.pcbi.1008215>, https://doi.org/10.1162/jocn_a_01914, <https://www.biorxiv.org/content/10.1101/2022.09.27.508760v1.full.pdf>, <https://doi.org/10.1111/nyas.14320>. This omission limits the authors' ability to situate their work within the existing scientific landscape and weakens the argument for their study's novelty and significance.

4. It would be good if the authors described the type of recurrent connectivity implemented in CORnet-S, and commented on how that differs from what we know from recurrent processing in the brain (recurrence within an area, with added skip connections).

5. Could the authors include a comment on the fact that they rely upon a single DNN instance per architecture to draw their conclusions? As shown by Mehrer et al (2020, Nature Communications), different network seeds give rise to considerable variance in the network internal representations.

6. Have the authors considered also looking into behavior (from both the network and the human participants)? Given that behavioral data was collected during the experiments, I think it would significantly enhance the depth and relevance of the paper if this data were included in the analysis. This could provide a direct connection between the underlying neural processes and observable human behavior, serving as a robust validation tool for the computational model. Further, it could augment the generalizability of the findings by demonstrating their manifestation not just at the neural level, but also at the behavioral level. Lastly, incorporating behavioral data could facilitate more confident causal inferences about the mechanisms involved in object recognition.

Related, could the authors comment on how they think the task demands influenced their results?

Reviewer #3 (Remarks to the Author):

This paper reports a set of representational similarity analyses on previously published fMRI and MEG data to investigate the spatiotemporal dynamics of semantic information representation in the human brain, disentangled from visual properties of images reflected in artificial neural network model (CorNet-S) activations.

The two aims of the paper, (1) contrasting the influence of visual vs. semantic information, and (2) establishing the time course and 'direction' of object information processing through the brain, are interesting and worthwhile. The proposed insight offered by this paper, namely that 'feedback activity to the posterior ventral lobe is important for object recognition, and feedback within the temporal lobe is most relevant to the processing of higher-level visual and semantic object properties between 200-

500 ms' (P9) would be a valuable contribution to the literature. Unfortunately, I am not convinced that the current manuscript provides solid support for these claims. This is due to several weaknesses:

- The paper in current form is not so easy to understand, mostly because information is not optimally presented: the Intro is quite broad, important methodological details are missing (e.g., what object classes were presented, what semantic features were measured) or poorly explained (e.g. the logic of the RCA analysis), and it's hard to understand the Results before reading the Methods (the order in which the paper is currently presented).

- Some of the wording used overstates the results. For example: 'we show that recurrent activity supports object recognition in the brain' (p9): the results show that recurrent activity (if this is indeed what the RCA analysis can disambiguate, see next point) is present when subject name images of objects, not that it is critical or important to recognize those images; concluding that requires a link with behaviour. The lack of measurements or analysis of behaviour makes the impact of the reported brain patterns on object recognition unclear.

- As far as I know the RCA method is not a widely used and established method (as opposed to, for example, temporal generalization analysis that is often used in decoding style analysis; see Contini et al., 2017, for a review <https://doi.org/10.1016/j.neuropsychologia.2017.02.013>), and I feel it needs more motivation and validation to support the current claims about information processing in the brain. Why is it valid to assume that partialing out time points prior to time t of the 'posterior half' of the sensors from the RSA correlation reflects feed-forward information processing, while partialing out the 'anterior half' reflects feedback? This seems a rather crude assessment of feedforward vs feedback, and to assume that all feedback is coming from frontal cortex?. The paper mentions one prior study that introduced this technique – did they validate the method and explain its underlying assumptions?

- One curious aspect of the MEG results is that many of the reported effects extend beyond the stimulus presentation period, which made me wonder about the influence of image offsets and response requirements on the suggested feedforward/feedback patterns. Offset responses to images are common in visual cortex neurons, showing re-activation with the disappearance of the image – could this contribute to the ongoing feed-forward effects beyond the 500 ms stimulus duration? And what was the average latency in the naming task that the subjects were performing – could response preparation contribute to the proposed feedback effects? (Again behavior could be informative here).

Below, I provide more detailed comments reflecting comments and questions I had while reading the paper. I hope these will be helpful to the authors to improve the paper moving forward.

Introduction:

The writing here is quite 'high-level', highlighting several times a need to understand the dynamics of representation of visual vs. semantic features, without giving a clear explanation or examples of what is meant with semantics or detailing specific semantic models and hypotheses that will be tested. It also contains a few typo's. Below are a few examples where the Intro could be improved:

- 'Critical to separate semantic vs visual', 'critical new domain', 'critical issue of understanding the relationships between visual and semantic processing' -> critical for what?

- What is meant with semantics/ meaning of the item, as opposed to 'simply discerning a label'? It would be helpful to have examples.

- Last sentence is ungrammatical.

- Figure 1 is referred to in a sentence that talks about 'a variety of neuroimaging approaches' to a 'large variety of object categories', but they figure only shows different types of MEG analyses and 2 objects.

Results:

General remarks:

- A motivation to use the CORnet-S should be outlined in Intro/Results, along with a discussion of how its accuracy on image recognition benchmarks is taken into account, as there are certainly better performing ANNs than the CorNet family, which could potentially incorporate more semantics. Can we draw such broad conclusions about visual features vs. semantics from testing a single ANN model?
- The semantic feature model should be explained in Intro/Results, and possibly a figure. (The sentence "The semantic model is created using semantic features" caption Fig 1, is not so informative).
- 'RSA model fit' is not the right term to use, what is reported are (partial) correlations; the use of 'model fit' suggests some kind of optimisation procedure as is common in encoding model approaches that use cross-validation or testing on on held out test data, all of which is not applied here as far as I can tell.

fMRI RSA searchlight section:

- The text says the analysis 'replicates Clarke & Tyler 2014', but the analysis is based on the same data, so it seems incorrect to call it a replication. It would be good provide more detail on how the current analyses or models used are different from the original analyses in the paper the data is from.
- The section also lacks a clear conclusion. Why is this fMRI analysis included in the first place? Its interpretation (based on the information provided in the Introduction) is not clear.
- Page 4, what is 'voxel PS'?
- Figure 2: showing one hemisphere upside down is confusing imo. Caption B refers to 'model RDM in A', but A shows no RDMs.

MEG section:

- Fig 3:
 - A. As in Fig 2, it would be helpful to also show the 'regular' RSA correlations (not just the partial ones). Based on the current plot in 2A, it seems strange that there is so little unique correlation of CorNet-V2, but this could be because it shares a lot of variance with CorNet-V1, for example.
 - B. I'm not so sure what to make of these two plots, which seem to show the same data but referenced to another different RDM (IT vs semantic): is the point that the difference between CorNet-IT and semantic features in terms of onset is not significantly different? It would be helpful to indicate that explicitly (e.g., with a 'significance star'). And why do the distributions look different in shape when taking the CorNet-IT vs. the semantic model as reference?
- Fig 4: Again here the text claims to show some significant effects in panel C but this is not clearly indicated in the figure. The text also doesn't report a p-value (or percentage of the CI distribution) for the significant differences.
- Page 7 'concomitant feedback signals [...] pointing to the importance of recurrent activity': these results don't show the recurrent activity is important, they just show it is present. To draw a conclusion about importance, it would be necessary to show that eliminating this recurrent component leads to impaired object recognition, for example.
- Fig 5: It's confusing to discuss panel B before panel A. Panel C and D are not referenced in the text.
- Page 7 '[...] accessing semantic information about objects': I'm not sure if it is valid to conclude that these model correlations mean that semantic information was 'accessed'; the subjects were just naming the objects, perhaps this data is suggestive that they activated some semantic properties while doing so, but the process that underlies this is not clear from these data.

- Page 8 'with spatial distribution of semantic effects closely overlapping those revealed through fMRI RSA': it's quite hard to assess if this is accurate from the few (small) brain pictures shown. A claim like this should be supported by a quantitative analysis of the overlap.

Multivariate RCA analysis (Page 8 & Figure 6):

- I had trouble understand this analysis. In my understanding whether an MEG-RSA correlation could be considered 'feedforward' or 'feedback' is based on whether it is partialing out the time-points preceding current time t anterior vs. posterior sensors (Fig 1). How is feedforward determined using these clusters?

- While the results seem somewhat consistent across panels, here I also started to feel a bit uncomfortable with the extremely small correlation values on the y-axis. I understand these are partial correlations but still, how meaningful are effects of this size?

- There also seems to be potential circularity in the analysis, given that the clusters are first identified based on whether they show a unique partial correlation with semantic vs. ANN features, and then evaluated again for significance on this correlation in the plots in Fig. 6.

Methods:

- The model specification should include what data CorNet was trained on.

- MEG data: How many images/trials in total? How many repeats?

- Peak latency analysis: I don't understand how the n-leave-out is used in the determining the significance. The paragraph basically describes a permutation analysis, but doesn't talk about how the left-out data is used in this context.

- What software was used for these analyses? Matlab, Python, any pre-existing toolboxes? Is the data publicly available?

- The searchlight RSA section doesn't explicitly say this analyses was done in the source-reconstructed data. The source reconstruction itself is only very minimally described in MEG preprocessing section, I almost missed it.

- fMRI experiment description: seems more logical to start Methods with this since this is first reported in results. I found it hard to follow the details of the methods with the minimal information provided. For example, what is 'the grey matter mask' that was used to mask the t-maps; how was it obtained? Why were 'unnormalised EPI images' used to create the t-maps; unnormalized in what way? Unlike the MEG section, the fMRI section also lacks detail on how the RDM-correlations were obtained.

- Sensor RCA section 'in contrast to RSA analysis, upper triangle of each RDM was converted to an RDV'; usually in RSA also only the upper (or lower) triangle is used when comparing RDMs, this is important because including the whole matrix and/or the diagonal can inflate correlations (see Ritchie et al., 2017, <https://doi.org/10.1016/j.neuroimage.2016.12.079>).

Reviewer #1 (Remarks to the Author):

1. In general, how do you explain the ~150ms lag between the peak time of the first three layers of the CORnet and the CORnet-IT? It seems pretty long, in macaque, usually IT and V4 has about 50ms latency.

When looking at peak times for the MEG RSA analysis (Figure 3), there is indeed a ~150 ms lag between the peak times for the IT-layer and the preceding CORnet layers. This analysis takes into account the whole brain, so the peak effects are contributed to by many regions which each have different underlying dynamics – which might be what we see with the rise of IT effects that are slightly delayed compared to the rise of effects for previous layers. Therefore, it's possible that much smaller differences in peak latencies emerge if specific regions are targeted, such as cortical V4 and IT.

Perhaps an aspect of the data that better speaks to the shorter latency difference you mention can be seen in the differences in onset times of the significant clusters. While the significant time-windows begin ~70-100 ms in layers V1,V2 and V4, the time window begins at ~130 ms for IT giving at 30-60 ms latency difference. While we can't make inferences based on the onset times of significant clusters (see Sassenhagen & Draschow, 2019), it at least points towards a possible shorter delay in the detection of IT-like representations and V4-like representations. However, we didn't statistically assess onset times, instead focussing on peaks.

2. In general, semantic features has peak correlation with the brain data after ~300ms. Humans are able to recognize objects that are briefly presented (~50ms), meaning it does not always take 300ms to “access semantic memory.” Is this the nature of MEG data?

Yes we agree that images can be presented very briefly and still recognised, however the briefly presented images will still result in dynamic neural responses that will last hundreds of milliseconds, and can be seen in MEG, EEG and human intracranial recordings. For example, Quiñones Quiroga et al (2008) show single neurons in the MTL respond with similar durations across a number of presentation durations (33ms, 66ms, 132ms, 264ms) with firing rates being maximal ~300-500 ms. Another example would be our RSA analysis of ECOG signals in the ATL which found peak semantic effects between 200 and 500 ms (Clarke, 2020). These timings of semantic effects also align with studies of language which have linked neural responses in the 300-500 ms window (the N400 in EEG studies) to semantic access (reviewed in Kutas & Federmeier, 2011).

While this is the case for our task where participants need to access the specific name of the concept, we do acknowledge that other semantic tasks can be accomplished quicker, based on less specific semantic information. For example, both computational (Rogers & Patterson, 2007) and behavioural (Mace et al., 2009) works suggests that coarse or superordinate category representations can emerge rapidly before basic-level concepts, meaning that semantic tasks that do not require object-specific or basic-level names can be performed faster than ones that require a basic-level name. This is something we've explored and expanded upon elsewhere (e.g. Taylor et al., 2012; Clarke & Tyler, 2015; Clarke, 2019). This means that the task participants are asked to do can have a significant impact on the neural dynamics during object recognition (e.g. Clarke et al., 2011, JOCN).

3. The study is very similar to a study published from the Kriegeskorte group (Jozwik et al. 2023) except that the current study tries to disambiguate between the contribution of the feedforward and feedback processing. The paper that I cite here also compares CORnet-RT to brain data. It seems natural to try this model as well. There are some differences in the peak latency between the studies.

How do you explain the discrepancy? Also, in order to claim that semantic features explains beyond the vanilla object recognition model, it would be better to test whether including the semantic model actually explains more of the variance in the brain data. Jozwik et al. does this by fitting the brain data to different GLM models (one with both CORnet and semantic model and the other with just the CORnet).

Thank for pointing us to this recent paper, and you rightly highlight our fundamental contribution to disambiguate between the contribution of the feedforward and feedback processing, which is very distinct from the Jozwik paper (and others we have previously published using ANNs and semantic models together; Clarke et al., 2018; Clarke et al., 2015). We see that Jozwik et al. used both a feedforward version of CORnet (CORnet-Z) and a locally-recurrent version (CORnet-R, although they may have used the corrected CORnet-RT version). In their analysis, they combine these versions of CORnet and do not analyse the individual layers/areas, or separate the CORnet versions to see which is a better model of the data. However, following your suggestion, we have also tried part of our analysis with CORnet-RT (and other ANN models) to see if it can better capture semantic effects. For the MEG sensor RSA analysis, we have added a supplementary analysis which shows what variance the semantic feature model can explain after separately accounting for CORnet-S, CORnet-RT, Alexnet, Resnet50 and VGG19. We find semantic effects with similar timings when controlling for each ANN, but that semantic effects are most reduced when using CORnet-S. This suggests that CORnet-S is the ANN which provides the most compelling case that the semantic effects we see transcend those of vision. This new analysis can be seen in Supplementary Figure S2, and below:

Figure S2. RSA results for the semantic feature model after different ANNs have been partialled out. Shaded areas show standard error of the mean and dotted line shows the non-partial RSA results for the semantic feature model.

You also note that there are differences in the peak latencies between our results and that of Jozwik. While they do not report peak latencies, we can see that our results and theirs are in broad agreement for peak effects of the ANN. As they do not report layer-specific analyses, the comparison is tricky, but the latencies are broadly in line with our current results (for CORnet V1-V4), and those we have previously reported for different ANNs (Alexnet – Clarke et al., 2018; HMax – Clarke et al., 2015). All of these seem to peak between 80-120 ms. In terms of the semantic models, our peak effects occur in the 200 – 400 ms period, while Jozwik use a visuo-semantic model which peaks approximately in the 150-250 ms window. While it is hard to know if the peaks are different (since they don't report peaks, and those that can be seen might fall within our 95% CIs), we believe the potentially earlier effects in

Jozwik relate to how the semantic model is constructed, the task participants are engaged in, and the high number of repetitions of the item.

It is well known that neural responses to a repetition of the stimulus leads to a changed response, with the biggest change being between the first and second presentation. For example, during semantic processing, EEG signals reduce in amplitude between first and second repetition, while the latency of semantic effects reduces with repeated presentation (Renoult et al., 2012). In their study, Jozwik et al repeat each of the stimuli 20-28 times, while we only show the images once. Therefore, this may go some way to explain a shift in the latency of the semantic effects. A more significant factor is likely the differences in the way the semantic models are constructed. Jozwik et al construct a visual/semantic model, where the predictors are either visual features of the images (e.g. black, eye, head, tail etc) or category labels (e.g. giraffe, food, kiwi etc), with a large proportion of the predictors relating to a single image (see Figure 4 and 5 in Jozwik et al., 2016). The consequence is that the predictors will be tuned to pick out a visual feature of the image, or to be able to identify that image as different from the others.

The semantic model we use, does not contain any category labels, or taxonomic information as these kinds of features refer to a superordinate categories and are not normally regarded as true semantic features in studies of conceptual representation. Nor are basic-level labels considered features. The advantage of distributed models of conceptual knowledge, is that they allow you to create statistical models of concepts which are informative about what the concept is, and how it relates to other concepts. This can't be achieved with semantic models which are largely based on basic and category labels. Finally, we note that many of the images and predictors of the model relate to faces, which may explain some of the differences, and we purposefully do not include faces.

Some of this is now added to the discussion on page 15, and below:

... This relationship between visual and semantic effects is similar to our previously reported work (Clarke et al., 2015, 2018) and a recent study by Jozwik et al., (2023). While there is broad consistency in the timing of peak effects for the ANNs, there is more variability comparing our semantic feature effects to their visuo-semantic model. The MEG RSA results Jozwik et al., (2023) present show earlier peak effects for a visuo-semantic model, approximately 150-250 ms, compared to our peak semantic effects between 200 and 400 ms, in line with a large body of work on N400 semantic effects (see Kutas & Federmeier, 2011). The visuo-semantic model Jozwik used includes various visual features of colour, texture along with category labels (e.g. giraffe, food, kiwi) for the object images which may create an earlier shift in time. The semantic model we use is a relational model, which does not include any category labels or taxonomic information as these are not usually considered true semantic features in models of conceptual representations. Our semantic feature model, based on over 3000 features given to concepts presented as words, captures the similarities and differences between a broad set of concepts and is likely more abstracted from the visual images, contributing to the delayed model correlations. A final factor additionally contributing to the earlier visuo-semantic effects is the large number of stimuli repetitions used by Jozwik, compared to our single presentation. Semantic effects are known to change in amplitude and latency upon repetition (e.g. Renoult et al., 2012) although this is likely to influence visual effects less so.

Finally, you raise a point that the methodological approaches used by Jozwik and ourselves differ. Conceptually, they both aim to establish the same thing – is there a unique contribution of the different variables. In our approach, we can establish the correlation between the semantic feature effects and the neural data, over and above that explained by the visual measures. So, we are showing that the semantic feature model explains variance in the data beyond the what the visual measures can capture. Of course, the approach by Jozwik also does this, however, they lose the sign of the

relationship between the predictor and the outcome measures of the GLM (values will always be positive) meaning they lose the ability to see the nature of the relationship between the brain and model. We think keeping this is helpful and provides a more intuitive assessment between the models and the brain.

4. Are there images that elicited higher feedback signals more than other images? If so, are there any image properties that predict the engagement of feedback?

Thank you for this comment, which highlights what we think is a very important issue. Indeed, we believe that certain concepts would illicit more feedback than others. For example, when considering the ATL and pVTC, our prediction is that concepts that are more semantically confusable will engage increased feedback from the ATL to pVTC. While we don't know of an approach that will allow us to track connectivity for single trials, we can assess this by creating different sub-groups of our stimuli, and then comparing RCA effects for high and low groups. These groups can be created for high and low semantically confusable items, or those with more or fewer lexical competitors, or those that are visual more or less complex. This is the aim of our ongoing analyses, but we think this is beyond the scope of the work we present here.

One extra analysis we have now added to the manuscript, looks at how connectivity is modulated not at the image-level, but at the participant level. We now show that feedback between right ATL and pVTC is greatest for participants that take longer to name the images. This suggests that semantic feedback representations were useful for behaviour, consistent with claims feedback will occur until the concepts can be uniquely identified. This can be seen on page 12-13, and included below:

Feedback semantic feature effects are related to behavioural response latencies

Whilst we have shown that recurrent connectivity relates to semantic feature effects, most strongly between the pVTC and right ATL, this does not indicate whether these connectivity dynamics are behaviourally relevant. We reasoned that if this connectivity was behaviourally relevant, then changes in connectivity across participants should relate to changes in mean response times across participants. To assess this, we tested if RCA effect sizes varied according to reaction times. We used a median-split of our participants to create one group of 15 with faster mean response times, and another group of 15 with slower mean response times (overall mean naming latency 951 ms, st dev 120 ms, range 793-1169 ms). This allowed us to determine if feedforward or feedback RCA effects significantly varied depending on behaviour.

Contrasting RCA effects between these two groups of participants revealed that participants with slower responses displayed significantly greater feedback RCA than those with faster responses. This was observed between the right ATL and the pVTC for the semantic model RDM (time window = 384-420 ms, cluster $p = 0.0386$; Figure 8). These effects overlap in time with the feedback RCA effects of semantics seen for the whole group. There were no further behavioural effects seen for feedforward or feedback RCA. Additional exploration of the CORnet model layers also revealed no significant modulations of RCA by behaviour.

Figure 8. Relationship of RCA effects to behaviour. A). Feedforward RCA effects of the semantic model for the faster group (light grey) and slower group (dark grey). B). Feedback RCA effects for the two groups, where feedback is significantly increased for the slower group. Shaded areas show standard error of the mean. Solid bar show time periods of significant effects. C). Correlation plot showing the relationship between response time and RCA for each participant, and the fitted linear effect.

And in the discussion on page 14:

We also saw that participants who were slower at responding, showed comparably increased feedback activity, which might suggest that such tasks can be performed with minimal feedback if speeded responses are required (Rogers & Patterson, 2007). Further studies will be needed to more directly assess how the degree of feedback varies across individual concepts, image manipulations and tasks. For instance, we would predict that more semantically confusable items would require more feedback between the ATL and pVTC.

5. Are there any differences in brain signals for the incorrectly labeled trials? Either these mistakes are from perceptual or semantic errors. It would be interesting to see if you can see this in the brain data. For example, if these were perceptual errors (subjects didn't see them, or not paying attention could look at eye movement patterns) then the perceptual areas in the brain could have been less active. Also, you can compute if either the feedforward or the feedback processing is affected.

This is again a great suggestion, but one we can't implement here. There are unfortunately too few incorrectly named trials in our study, as the items are all common objects that were selected to be known to most people. But we agree it would be interesting to see how naming errors of specific types might be linked to differences in connectivity dynamics, and have been considering how to do this using backwards masking procedures.

We hope that the changes we have made, and our responses here, have satisfied you, and thank you for your considered review of our work.

References

- Clarke, A. (2019). Neural dynamics of visual and semantic object processing. In *Psychology of learning and motivation* (Vol. 70, pp. 71-95). Academic Press.
- Clarke, A., Taylor, K. I., & Tyler, L. K. (2011). The evolution of meaning: spatio-temporal dynamics of visual object recognition. *Journal of cognitive neuroscience*, 23(8), 1887-1899.

- Clarke, A., & Tyler, L. K. (2015). Understanding what we see: how we derive meaning from vision. *Trends in cognitive sciences*, 19(11), 677-687.
- Clarke, A. (2020). Dynamic activity patterns in the anterior temporal lobe represents object semantics. *Cognitive neuroscience*, 11(3), 111-121.
- Kutas, M., & Federmeier, K. D. (2011). Thirty years and counting: finding meaning in the N400 component of the event-related brain potential (ERP). *Annual review of psychology*, 62, 621-647.
- Macé, M. J. M., Joubert, O. R., Nespoulous, J. L., & Fabre-Thorpe, M. (2009). The time-course of visual categorizations: you spot the animal faster than the bird. *PLoS one*, 4(6), e5927.
- Quiroga, R. Q., Mukamel, R., Isham, E. A., Malach, R., & Fried, I. (2008). Human single-neuron responses at the threshold of conscious recognition. *Proceedings of the National Academy of Sciences*, 105(9), 3599-3604.
- Renoult, L., Wang, X., Calcagno, V., Prévost, M., & Debrulle, J. B. (2012). From N400 to N300: Variations in the timing of semantic processing with repetition. *NeuroImage*, 61(1), 206-215.
- Rogers, T. T., & Patterson, K. (2007). Object categorization: reversals and explanations of the basic-level advantage. *Journal of Experimental Psychology: General*, 136(3), 451.
- Sassenhagen, J, Draschkow, D. Cluster-based permutation tests of MEG/EEG data do not establish significance of effect latency or location. *Psychophysiology*. 2019; 56:e13335.
<https://doi.org/10.1111/psyp.13335>
- Taylor, K. I., Devereux, B. J., Acres, K., Randall, B., & Tyler, L. K. (2012). Contrasting effects of feature-based statistics on the categorisation and basic-level identification of visual objects. *Cognition*, 122(3), 363-374.

Reviewer #2 (Remarks to the Author):

In the paper "Recurrent connectivity supports higher-level visual and semantic object representations in the brain", von Seth et al. use neuroimaging and computational modelling to investigate the spatiotemporal dynamics during object recognition.

The authors differentiate between semantic and visual processing (processing the visual attributes of the image vs. semantic object processing). They investigate how the brain's representation of an object changes over time, and how that relates to both the visual and semantic aspects of an object. This paper presents an ambitious attempt to elucidate the neural mechanisms underlying visual object recognition, emphasizing the interplay between visual and semantic properties and the role of feedforward and feedback signals. The methodological approach is commendable, using techniques like Representational Similarity Analysis (RSA), Recurrent Connectivity Analysis (RCA), and implementing Artificial Neural Networks (ANN).

The topic is timely and of interest to the community. However, I believe the study's core contributions and aims are obfuscated due to the varying angles of introduction and unclear research question, which limits the overall impact of the paper. Below I highlight a few concerns that, I hope, will help the authors further strengthen their submission.

We'd like to thank you for your considered and helpful comments on the manuscript, and are glad you believe the study is interesting. We have now implemented changes that we hope address your concerns and the issues you raised, and believe the manuscript is much improved as a consequence.

Specific Comments:

1. I found it difficult to understand what the aim or research question of the study was, as the authors introduce the topic from different angles. This makes it challenging to evaluate the impact of the current findings.

I believe the authors have made interesting contributions, in showing for example that recurrent activity was linked to higher-level visual and semantic properties and was mainly driven by interactions between the anterior and posterior ventral temporal cortex. The research also highlighted the importance of semantic object information in explaining neural activity, thereby proving that semantic contributions go beyond those explained by ANNs alone (and it suggests a more dynamic and interactive model of semantic activation than perhaps previously proposed). However, in the current form I believe the core findings and contributions are obfuscated and it is for the reader not immediately clear what to do with this information or how this goes beyond what we already know from previous literature.

We're sorry that the main aims and contributions of the study were not stated as clearly as they could have been in the original paper. We're dealing with multiple ideas and techniques, and it is now clear they were not communicated in the best way. We have since made substantial changes to the manuscript to address this, including making the main aims of the study more explicit throughout the introduction, and referencing them again in the results. We have also expanded our descriptions of the connectivity approach, which emphasises its importance to the research. To make the aims clearer, we have made changes on page 2, and include these below:

While prior studies have shed some light on how semantic object processing relates to, and is distinct from, processing the physical visual attributes of the image (Bankson et al., 2018; Clarke et al., 2013, 2015, 2018; Jozwik et al., 2023; Rupp et al., 2017), in this study we look to probe the evolving nature of object representations in a new domain, asking how dynamic connectivity patterns relate to the visual and semantic aspects of objects. Using a representational connectivity approach (Karimi-Rouzbahani et al., 2022) will allow us to reveal the core object properties that feedforward and feedback connectivity relate to, helping to shape cognitive accounts of how we understand the meaning of what we see.

And on page 3:

However, it remains unclear exactly which aspects of visual and semantic object information relate to feedforward and feedback signals when recognising unambiguous familiar objects. To address this, we use a recently developed method for model-based representational connectivity analysis (RCA; Karimi-Rouzbahani et al., 2021, 2022), which builds on prior examples of understanding connectivity using representational similarity approaches for fMRI (Anzellotti & Coutanche, 2018; Clarke et al., 2022; Kriegeskorte, 2008; Pillet et al., 2020). In the case of time-resolved data, RCA can show how activity in one region contributes to information-specific activity in another region at a later point in time, with the potential to reveal how feedforward and feedback signals relate to visual and semantic information. This follows the recent development of multivariate informational connectivity approaches (Anzellotti & Coutanche, 2018; Basti et al., 2020; Goddard et al., 2016; Rahimi, Jackson, et al., 2022), which seek to provide information about the timing and direction of between-region connectivity, alongside pointing towards the representational content connectivity can support. To infer timing and direction, RCA uses similar principles to Granger Causality, in that it evaluates whether past information in one region can help explain the current representational patterns in another region. While this kind of approach has been used in a number of studies to evaluate if there is shared information between regions over time (e.g. Goddard et al., 2022; Kietzmann et al., 2019; Lyu et al., 2019), the RCA approach we employ here goes one step further by specifying the precise nature of that shared information (Karimi-Rouzbahani et al., 2021, 2022) – namely, is connectivity helping shape visual or semantic representations.

And page 4:

Here, we address two overarching issues relating to, first, the role of feedforward and feedback signals during objects recognition, and second the relationship between visual and semantic object properties. To achieve this, we use both fMRI and MEG with representational similarity analysis (RSA; Kriegeskorte, 2008; Nili et al., 2014) (Figure 1) and MEG-RCA applied to the recognition of visual objects from a large variety of object categories. Building across complimentary analyses we (1) use fMRI searchlight RSA to reveal the cortical architecture related to semantic object properties and how they relate to those explained by a computational model of vision; (2) explore the relative temporal dynamics and connectivity of visual and semantic measures at the level of MEG sensor arrays using both RSA and RCA; and (3) examine the spatio-temporal distribution of semantic effects, beyond those explained by a computation model of vision, using searchlight RSA of MEG source localised neural patterns. Finally we (4) test how connectivity within the resulting network relates to semantics and visual object properties. Together these analyses illustrate how visual and semantic processes relate to one another over time and space during object recognition. Importantly, these analyses shed light on the visual and semantic representations that might be supported by feedforward and feedback signals across regions along the VVP.

Finally, although the discussion currently emphasises how semantic cognition is underpinned by a dynamic neural system, we have now also added new information on the implications of these results for future ANNs, on page 15/16:

The current work has implications for future developments of brain inspired ANNs. Our work aligns with computational work suggesting ANNs are improved by between-layer feedback connections in addition to the within-area recurrent (or lateral) and skip connections (Spoerer et al., 2017). For models such as CORnet, which makes links between network areas and neural regions, our work suggests that they would more accurately account for object recognition through the addition of an 'ATL' layer with nodes trained to reflect conceptual structure (as in Devereux et al., 2018), and with feedback from the ATL to other layers (see O'Reilly et al., 2013). Such extensions could also track the stimulus history to enable models to account for priming effects and MTL-like behaviours (Bonnen et al., 2021).

We hope these changes make it clear the central aim of understanding the contributions of feedforward and feedback connectivity.

2. Related, I found the paper to be missing references to and a discussion of previous experimental and computational work from various labs that investigate the functional role of recurrent computations during object recognition (or the incorporation of semantic features). To list a few...:

This omission limits the authors' ability to situate their work within the existing scientific landscape and weakens the argument for their study's novelty and significance.

Thank you for pointing us to these works, and have now added reference to them in appropriate places. We have also added a discussion of the relationship of the recent Jozwik paper to our own in the discussion on page 15, and comments on the work from Spoener on page 15 (and elsewhere):

This relationship between visual and semantic effects is similar to our previously reported work (Clarke et al., 2015, 2018) and a recent study by Jozwik et al., (2023). While there is broad consistency in the timing of peak effects for the ANNs, there is more variability comparing our semantic feature effects to their visuo-semantic model. The MEG RSA results Jozwik et al., (2023) present show earlier peak effects for a visuo-semantic model, approximately 150-250 ms, compared to our peak semantic effects between 200 and 400 ms, in line with a large body of work on N400 semantic effects (see Kutas & Federmeier, 2011). The visuo-semantic model Jozwik used includes various visual features of colour, texture along with category labels (e.g. giraffe, food, kiwi) for the object images which may create an earlier shift in time. The semantic model we use is a relational model, which does not include any category labels or taxonomic information as these are not usually considered true semantic features in models of conceptual representations. Our semantic feature model, based on over 3000 features given to concepts presented as words, captures the similarities and differences between a broad set of concepts and is likely more abstracted from the visual images, contributing to the delayed model correlations. A final factor additionally contributing to the earlier visuo-semantic effects is the large number of stimuli repetitions used by Jozwik, compared to our single presentation. Semantic effects are known to change in amplitude and latency upon repetition (e.g. Renoult et al., 2012) although this is likely to influence visual effects less so.

Page 15:

Our work aligns with computational work suggesting ANNs are improved by between-layer feedback connections in addition to the within-area recurrent (or lateral) and skip connections (Spoerer et al., 2017). For models such as CORnet, which makes links between network areas and neural regions, our

work suggests that they would more accurately account for object recognition through the addition of an 'ATL' layer with nodes trained to reflect conceptual structure (as in Devereux et al., 2018), and with feedback from the ATL to other layers (see O'Reilly et al., 2013).

3. It would be good if the authors described the type of recurrent connectivity implemented in CORnet-S, and commented on how that differs from what we know from recurrent processing in the brain (recurrence within an area, with added skip connections).

We have now added a description of CORnet-S to the beginning of the results section, and emphasised that it involves locally-recurrent circuits with no between area connections. We believe there was some ambiguity in the previous manuscript where both CORnet and our results were referred to as recurrent, which did not emphasise that CORnet involves no between area connections, while our results are exclusively about between area connections. We have now made sure to refer to CORnet as a locally-recurrent model, and emphasised that we are analysing between region recurrent connectivity.

The description of CORnet-S is on page 4 and below:

We used a pre-trained version of CORnet-S to obtain visual measures of our stimuli (extracted from Muttenthaler & Hebart, 2021). CORnet-S is composed of different processing units which are conceptualised as capturing the visual areas V1, V2, V4 and IT of the non-human primate brain (Kubilius et al., 2019). CORnet-S has locally recurrent processing within each area (with no between area feedback), and is amongst the best performing models in predicting neural responses recorded in macaque IT, and performs well in capturing human behavioural similarity judgements (Kar et al., 2019; Kubilius et al., 2019). Nodal activations for each area of CORnet-S were extracted from the area's convolutional layer. From these, we calculated representational dissimilarity matrices (RDMs) that quantify how similar or different the activations of the layer were between all the images.

We have also emphasised how we believe ANNs could be improved by including between area connections in future instantiations, as mentioned above, on page 15/16, and included again below:

The current work has implications for future developments of brain inspired ANNs. Our work aligns with computational work suggesting ANNs are improved by between-layer feedback connections in addition to the within-area recurrent (or lateral) and skip connections (Spoerer et al., 2017). For models such as CORnet, which makes links between network areas and neural regions, our work suggests that they would more accurately account for object recognition through the addition of an 'ATL' layer with nodes trained to reflect conceptual structure (as in Devereux et al., 2018), and with feedback from the ATL to other layers (see O'Reilly et al., 2013). Such extensions could also track the stimulus history to enable models to account for priming effects and MTL-like behaviours (Bonnen et al., 2021).

4. Could the authors include a comment on the fact that they rely upon a single DNN instance per architecture to draw their conclusions? As shown by Mehrer et al (2020, Nature Communications), different network seeds give rise to considerable variance in the network internal representations.

We have now commented on this in the methods section, on page 17 and below:

Here we used a single pre-trained instance of CORnet-S, however, it has been suggested that nodal activations vary depending on the initial random seed of the network (Mehrer et al., 2020). While it is unclear if this variability would influence our RSA results (or those of other studies), we show the generalisability of our MEG semantic effects against a selection of different ANN architectures (see Supplementary Figure S2). Specifically, we also creating model RDMs for CORnet-RT (areas V1, V2, V4

and IT), AlexNet, Resnet50 and VGG19, where activations were extracted using THINGSVision (Muttenthaler & Hebart, 2021).

5. Have the authors considered also looking into behavior (from both the network and the human participants)? Given that behavioral data was collected during the experiments, I think it would significantly enhance the depth and relevance of the paper if this data were included in the analysis. This could provide a direct connection between the underlying neural processes and observable human behavior, serving as a robust validation tool for the computational model. Further, it could augment the generalizability of the findings by demonstrating their manifestation not just at the neural level, but also at the behavioral level. Lastly, incorporating behavioral data could facilitate more confident causal inferences about the mechanisms involved in object recognition. Related, could the authors comment on how they think the task demands influenced their results?

Thank you for this suggestion, as it was not something we'd previously considered. Our RCA analysis provides a single measure for each participant for each time-point, meaning we can only look at behaviour at the participant-level. The behavioural data we have is mean reaction times (naming latencies) for each participant. What we have now done, is created a median split of our participants giving a relatively faster group, and a slower group. We then compared the RCA results between the two groups for feedforward and feedback effects. This revealed that the slower RT group has significantly greater feedback between the right ATL and pVTC than the faster group. This suggests that semantic feedback representations were useful for behaviour, consistent with claims feedback will occur until the concepts can be uniquely identified, which is aided by feedback. This new result has now been added to the results section on page 12-13, and below:

Feedback semantic feature effects are related to behavioural response latencies

Whilst we have shown that recurrent connectivity relates to semantic feature effects, most strongly between the pVTC and right ATL, this does not indicate whether these connectivity dynamics are behaviourally relevant. We reasoned that if this connectivity was behaviourally relevant, then changes in connectivity across participants should relate to changes in mean response times across participants. To assess this, we tested if RCA effect sizes varied according to reaction times. We used a median-split of our participants to create one group of 15 with faster mean response times, and another group of 15 with slower mean response times (overall mean naming latency 951 ms, st dev 120 ms, range 793-1169 ms). This allowed us to determine if feedforward or feedback RCA effects significantly varied depending on behaviour.

Contrasting RCA effects between these two groups of participants revealed that participants with slower responses displayed significantly greater feedback RCA than those with faster responses. This was observed between the right ATL and the pVTC for the semantic model RDM (time window = 384-420 ms, cluster $p = 0.0386$; Figure 8). These effects overlap in time with the feedback RCA effects of semantics seen for the whole group. There were no further behavioural effects seen for feedforward or feedback RCA. Additional exploration of the CORnet model layers also revealed no significant modulations of RCA by behaviour.

Figure 8. Relationship of RCA effects to behaviour. A). Feedforward RCA effects of the semantic model for the faster group (light grey) and slower group (dark grey). B). Feedback RCA effects for the two groups, where feedback is significantly increased for the slower group. Shaded areas show standard error of the mean. Solid bar show time periods of significant effects. C). Correlation plot showing the relationship between response time and RCA for each participant, and the fitted linear effect.

And in the discussion on page 14:

Here, we revealed that high-level visual and semantic information are uniquely related to these dynamic recurrent interactions, and that feedback signals relating to semantic effects were modulated with behaviour.

...

We also saw that participants who were slower at responding, showed comparably increased feedback activity, which might suggest that such tasks can be performed with minimal feedback if speeded responses are required (Rogers & Patterson, 2007). Further studies will be needed to more directly assess how the degree of feedback varies across individual concepts, image manipulations and tasks. For instance, we would predict that more semantically confusable items would require more feedback between the ATL and pVTC.

We hope these changes have satisfied you, and thank you again for your considered review.

Reviewer #3 (Remarks to the Author):

This paper reports a set of representational similarity analyses on previously published fMRI and MEG data to investigate the spatiotemporal dynamics of semantic information representation in the human brain, disentangled from visual properties of images reflected in artificial neural network model (CorNet-S) activations.

The two aims of the paper, (1) contrasting the influence of visual vs. semantic information, and (2) establishing the time course and 'direction' of object information processing through the brain, are interesting and worthwhile. The proposed insight offered by this paper, namely that 'feedback activity to the posterior ventral lobe is important for object recognition, and feedback within the temporal lobe is most relevant to the processing of higher-level visual and semantic object properties between 200-500 ms' (P9) would be a valuable contribution to the literature. Unfortunately, I am not convinced that the current manuscript provides solid support for these claims. This is due to several weaknesses:

- The paper in current form is not so easy to understand, mostly because information is not optimally presented: the Intro is quite broad, important methodological details are missing (e.g., what object classes were presented, what semantic features were measured) or poorly explained (e.g. the logic of the RCA analysis), and it's hard to understand the Results before reading the Methods (the order in which the paper is currently presented).

We have now adjusted the introduction to make the main aims of the study more explicit, and added methodological details to the introduction, and to the beginning of the results section. We hope this gives the paper a better flow, without the necessity to skip to the methods before the results. We have also added extra explanation of the RCA analysis throughout the introduction and results (see response to your comment about this below), and explained the CORnet and semantic models (see responses below). We have also added details about the kinds of objects people viewed to the beginning of the results. We hope these changes lead to a more understandable paper.

To make the aims clearer, we have made changes to the introduction on page 2, and include these below:

While prior studies have shed some light on how semantic object processing relates to, and is distinct from, processing the physical visual attributes of the image (Bankson et al., 2018; Clarke et al., 2013, 2015, 2018; Jozwik et al., 2023; Rupp et al., 2017), in this study we look to probe the evolving nature of object representations in a new domain, asking how dynamic connectivity patterns relate to the visual and semantic aspects of objects. Using a representational connectivity approach (Karimi-Rouzbahani et al., 2022) will allow us to reveal the core object properties that feedforward and feedback connectivity relate to, helping to shape cognitive accounts of how we understand the meaning of what we see.

And on page 3:

However, it remains unclear exactly which aspects of visual and semantic object information relate to feedforward and feedback signals when recognising unambiguous familiar objects. To address this, we use a recently developed method for model-based representational connectivity analysis (RCA; Karimi-Rouzbahani et al., 2021, 2022), which builds on prior examples of understanding connectivity using representational similarity approaches for fMRI (Anzellotti & Coutanche, 2018; Clarke et al., 2022; Kriegeskorte, 2008; Pillet et al., 2020). In the case of time-resolved data, RCA can show how activity in

one region contributes to information-specific activity in another region at a later point in time, with the potential to reveal how feedforward and feedback signals relate to visual and semantic information. This follows the recent development of multivariate informational connectivity approaches (Anzellotti & Coutanche, 2018; Basti et al., 2020; Goddard et al., 2016; Rahimi, Jackson, et al., 2022), which seek to provide information about the timing and direction of between-region connectivity, alongside pointing towards the representational content connectivity can support. To infer timing and direction, RCA uses similar principles to Granger Causality, in that it evaluates whether past information in one region can help explain the current representational patterns in another region. While this kind of approach has been used in a number of studies to evaluate if there is shared information between regions over time (e.g. Goddard et al., 2022; Kietzmann et al., 2019; Lyu et al., 2019), the RCA approach we employ here goes one step further by specifying the precise nature of that shared information (Karimi-Rouzbahani et al., 2021, 2022) – namely, is connectivity helping shape visual or semantic representations.

And page 4:

Here, we address two overarching issues relating to, first, the role of feedforward and feedback signals during objects recognition, and second the relationship between visual and semantic object properties. To achieve this, we use both fMRI and MEG with representational similarity analysis (RSA; Kriegeskorte, 2008; Nili et al., 2014) (Figure 1) and MEG-RCA applied to the recognition of visual objects from a large variety of object categories. Building across complimentary analyses we (1) use fMRI searchlight RSA to reveal the cortical architecture related to semantic object properties and how they relate to those explained by a computational model of vision; (2) explore the relative temporal dynamics and connectivity of visual and semantic measures at the level of MEG sensor arrays using both RSA and RCA; and (3) examine the spatio-temporal distribution of semantic effects, beyond those explained by a computation model of vision, using searchlight RSA of MEG source localised neural patterns. Finally we (4) test how connectivity within the resulting network relates to semantics and visual object properties. Together these analyses illustrate how visual and semantic processes relate to one another over time and space during object recognition. Importantly, these analyses shed light on the visual and semantic representations that might be supported by feedforward and feedback signals across regions along the VVP.

Changes relating to the methods can be seen in the responses to the other points you raise below.

- Some of the wording used overstates the results. For example: ‘we show that recurrent activity supports object recognition in the brain’ (p9): the results show that recurrent activity (if this is indeed what the RCA analysis can disambiguate, see next point) is present when subject name images of objects, not that it is critical or important to recognize those images; concluding that requires a link with behaviour. The lack of measurements or analysis of behaviour makes the impact of the reported brain patterns on object recognition unclear.

We have now carefully gone through the paper to make sure we do not overstate effects. In response to your comment about behaviour, and those of reviewer 2, we have now conducted an analysis of the RCA effects in relation to behaviour. Our RCA analysis provides a single measure for each participant for each time-point, meaning we can look at behaviour at the participant-level. The behavioural data we have is mean reaction times (naming latencies) for each participant. What we have now done, is created a median split of our participants to create a relatively faster group, and a slower group. We then compared the RCA results between the two groups for feedforward and feedback effects. This revealed that the slower RT group has significantly greater feedback between the right ATL and pVTC than the faster group. This suggests that semantic feedback representations were useful for behaviour, consistent with claims feedback will occur until the concepts can be

uniquely identified, which is aided by feedback. This new result has now been added to the results section on page 12-13, and below:

Feedback semantic feature effects are related to behavioural response latencies

Whilst we have shown that recurrent connectivity relates to semantic feature effects, most strongly between the pVTC and right ATL, this does not indicate whether these connectivity dynamics are behaviourally relevant. We reasoned that if this connectivity was behaviourally relevant, then changes in connectivity across participants should relate to changes in mean response times across participants. To assess this, we tested if RCA effect sizes varied according to reaction times. We used a median-split of our participants to create one group of 15 with faster mean response times, and another group of 15 with slower mean response times (overall mean naming latency 951 ms, st dev 120 ms, range 793-1169 ms). This allowed us to determine if feedforward or feedback RCA effects significantly varied depending on behaviour.

Contrasting RCA effects between these two groups of participants revealed that participants with slower responses displayed significantly greater feedback RCA than those with faster responses. This was observed between the right ATL and the pVTC for the semantic model RDM (time window = 384-420 ms, cluster $p = 0.0386$; Figure 8). These effects overlap in time with the feedback RCA effects of semantics seen for the whole group. There were no further behavioural effects seen for feedforward or feedback RCA. Additional exploration of the CORnet model layers also revealed no significant modulations of RCA by behaviour.

Figure 8. Relationship of RCA effects to behaviour. A). Feedforward RCA effects of the semantic model for the faster group (light grey) and slower group (dark grey). B). Feedback RCA effects for the two groups, where feedback is significantly increased for the slower group. Shaded areas show standard error of the mean. Solid bar show time periods of significant effects. C). Correlation plot showing the relationship between response time and RCA for each participant, and the fitted linear effect.

And in the discussion on page 14:

Here, we revealed that high-level visual and semantic information are uniquely related to these dynamic recurrent interactions, and that feedback signals relating to semantic effects were modulated with behaviour.

...

We also saw that participants who were slower at responding, showed comparably increased feedback activity, which might suggest that such tasks can be performed with minimal feedback if speeded responses are required (Rogers & Patterson, 2007). Further studies will be needed to more directly

assess how the degree of feedback varies across individual concepts, image manipulations and tasks. For instance, we would predict that more semantically confusable items would require more feedback between the ATL and pVTC.

- As far as I know the RCA method is not a widely used and established method (as opposed to, for example, temporal generalization analysis that is often used in decoding style analysis; see Contini et al., 2017, for a review), and I feel it needs more motivation and validation to support the current claims about information processing in the brain. Why is it valid to assume that partialing out time points prior to time t of the ‘posterior half’ of the sensors from the RSA correlation reflects feedforward information processing, while partialing out the ‘anterior half’ reflects feedback? This seems a rather crude assessment of feedforward vs feedback, and to assume that all feedback is coming from frontal cortex?. The paper mentions one prior study that introduced this technique – did they validate the method and explain its underlying assumptions?

While the use of RCA incorporating model RDMs is less widely used, RCA has been applied in many previous fMRI, MEG and EEG studies in one form or another. The main concepts behind RCA were introduced in one of the early Kriegeskorte et al. papers (2008), and have since been used in fMRI (e.g. Clarke et al., 2022; Pillet et al., 2020; Coutanche et al., 2020), EEG (e.g. Goddard et al., 2016) and MEG (e.g. Kietzman et al., 2019; Lyu et al., 2020; Goddard et al., 2022). There are variations of the approach, but all are composed from the idea of correlating RDMs between regions. With MEG and EEG we can exploit time, allowing for inferences similar to Granger Causality about direction. These ideas have further been validated across different studies (Karimi-Rouzbahani et al., 2019; 2021) and in a more recent methods paper (Karimi-Rouzbahani et al., 2022). We have now added more context and information about RCA and its particular use, to the introduction on page 3 and below:

To address this, we use a recently developed method for model-based representational connectivity analysis (RCA; Karimi-Rouzbahani et al., 2021, 2022), which builds on prior examples of understanding connectivity using representational similarity approaches for fMRI (Anzellotti & Coutanche, 2018; Clarke et al., 2022; Kriegeskorte, 2008; Pillet et al., 2020). In the case of time-resolved data, RCA can show how activity in one region contributes to information-specific activity in another region at a later point in time, with the potential to reveal how feedforward and feedback signals relate to visual and semantic information. This follows the recent development of multivariate informational connectivity approaches (Anzellotti & Coutanche, 2018; Basti et al., 2020; Goddard et al., 2016; Rahimi, Jackson, et al., 2022), which seek to provide information about the timing and direction of between-region connectivity, alongside pointing towards the representational content connectivity can support. To infer timing and direction, RCA uses similar principles to Granger Causality, in that it evaluates whether past information in one region can help explain the current representational patterns in another region. While this kind of approach has been used in a number of studies to evaluate if there is shared information between regions over time (e.g. Goddard et al., 2022; Kietzmann et al., 2019; Lyu et al., 2019), the RCA approach we employ here goes one step further by specifying the precise nature of that shared information (Karimi-Rouzbahani et al., 2021, 2022) – namely, is connectivity helping shape visual or semantic representations.

In addition to expanding the descriptions of the approach throughout the results (see response to comments below concerning RCA).

- One curious aspect of the MEG results is that many of the reported effects extend beyond the stimulus presentation period, which made me wonder about the influence of image offsets and response requirements on the suggested feedforward/feedback patterns. Offset responses to images are common in visual cortex neurons, showing re-activation with the disappearance of the image –

could this contribute to the ongoing feed-forward effects beyond the 500 ms stimulus duration? And what was the average latency in the naming task that the subjects were performing – could response preparation contribute to the proposed feedback effects? (Again behavior could be informative here).

We agree it is likely that the feedforward RCA effects of V1 and V4 post 500 ms are likely due to a visual offset response to the picture. These can be seen in Figure 4, which takes into account all the sensors across the head. Interestingly, these late feedforward effects are not significant for the source clusters, which would be consistent with a visual offset response in primary visual areas, but not the higher-level visual regions covered by the clusters.

With respect the naming latencies, the mean response time was 951 ms, and beyond the range of our analysis epoch. However, as discussed above, the semantic feedback effects between pVTC and right ATL do relate to response times.

Below, I provide more detailed comments reflecting comments and questions I had while reading the paper. I hope these will be helpful to the authors to improve the paper moving forward.

Introduction:

The writing here is quite ‘high-level’, highlighting several times a need to understand the dynamics of representation of visual vs. semantic features, without giving a clear explanation or examples of what is meant with semantics or detailing specific semantic models and hypotheses that will be tested. It also contains a few typo’s. Below are a few examples where the Intro could be improved:

- ‘Critical to separate semantic vs visual’ , ‘critical new domain’, ‘critical issue of understanding the relationships between visual and semantic processing’ -> critical for what?

Thank you – we now see there was both an overuse of ‘critical’ and lack of explanation for what something was critical for achieving. This has now been changed, where ‘critical’ was either removed, replaced or expanded upon.

- What is meant with semantics/ meaning of the item, as opposed to ‘simply discerning a label’? It would be helpful to have examples.

We have now expanded on this point to better emphasize the distinction. Semantic memory for concepts, which includes the meaning of objects, encompasses our knowledge about and facts concerning those concepts. An essential aspect of semantic knowledge is that it is relational, in that we can use it to understand that bananas and apples share something in common despite being perceptually very different. While this is captured in semantic meaning, it is not within ANNs or output layers of ANNs that are composed of labels. They can identify the category of the image, but not how related two images are to one another in meaning. We understand that the goal of most ANNs is not to link perceptual and semantic knowledge, however, it is important to note that when naming an object, there are many cognitive processes between visual perception and the utterance of a name. Influential models of picture naming, for example, highlight that visual perception flows into semantic or conceptual processing, prior to lexical selection, and so we see that a natural extension of ANNs is to map onto conceptual representations rather than labels. We have now adjusted the statement on page 3, and added below:

While ANNs have shown an impressive ability to distinguish between different objects, they have done so by discerning a label for an image. However, this cannot capture how different semantic concepts relate to one another, such as apples and bananas having some shared meaning. If the goal of visual object recognition is to understand what the object being perceived is, then this requires semantic memory. An essential aspect of semantic knowledge is that it is relational. Semantic distances have long been thought to capture the underlying organisation of semantic memory (e.g. Rips et al., 1973), which can be used to understand that bananas and apples share something in common despite being perceptually very different.

- Last sentence is ungrammatical.

Thank you, this has now been corrected.

- Figure 1 is referred to in a sentence that talks about ‘a variety of neuroimaging approaches’ to a ‘large variety of object categories’, but they figure only shows different types of MEG analyses and 2 objects.

This has now been adjusted on page 4, and a new Figure 1, as shown below:

To achieve this, we use both fMRI and MEG with representational similarity analysis (RSA; Kriegeskorte, 2008; Nili et al., 2014) (Figure 1) and MEG-RCA applied to the recognition of visual objects from a large variety of object categories.

Figure 1. Schematic illustration of model and neural RDM construction for representational similarity and representational connectivity analyses. A) Construction of model RDMs. (i) 302 objects including animate and inanimate, natural and man-made objects were shown to participants and used to construct model RDMs. Each image and object was only shown once to each participant. (ii) Visual RDMs are created from pairwise comparisons of nodal activations extracted from 4 layers of the CORnet-S ANN for each object and vectorised. (iii) A semantic model RDM is created based on data from a large property norming study which generated

3026 features, with the RDM defined by the overlap in features between concepts, and vectorised. (iv) Pairwise Spearman's correlations between visual and semantic feature models show a high degree of correlation between the visual RDMs, graded by distance, but limited correlation between visual and semantic models. B) Construction of vectorised model RDMs from MEG and fMRI data. (i) fMRI searchlight RDMs reflect the similarity between voxel patterns for each of the objects, and each searchlight location across the brain. (ii) For the sensor-level MEG RSA analysis, MEG RDMs are created from object-specific spatio-temporal patterns for each time-point extracted from MEG sensors. (iii) Temporally resolved MEG RSA searchlight analysis is conducted for the semantic model using source localised MEG patterns. Vertices are illustrated with grey dots with shaded searchlight spheres, and the degree of hypothetical model correlation is indicated by purple colouration.

We have also now referred to the range and number of items at the start of the results section, which can be seen on page 4 and below:

In both fMRI and MEG datasets, participants viewed and named a range of visual images including different animals, foods, vegetables, fruits, vehicles, musical instruments, tools, clothing, and other common household objects (131 items in fMRI, 302 items in MEG). Using the responses to these individual concepts, we could then test whether brain-based similarity between the items related to our visual and semantic measures (see methods).

Results:

General remarks:

- A motivation to use the CORnet-S should be outlined in Intro/Results, along with a discussion of how its accuracy on image recognition benchmarks is taken into account, as there are certainly better performing ANNs than the CorNet family, which could potentially incorporate more semantics. Can we draw such broad conclusions about visual features vs. semantics from testing a single ANN model?

We have now added a description and motivation for CORnet-S to the results section as followings, and can be seen on page 4:

We used a pre-trained version of CORnet-S to obtain visual measures of our stimuli (extracted from Muttenthaler & Hebart, 2021). CORnet-S is composed of different processing units which are conceptualised as capturing the visual areas V1, V2, V4 and IT of the non-human primate brain (Kubilius et al., 2019). CORnet-S has locally recurrent processing within each area (with no between area feedback), and is amongst the best performing models in predicting neural responses recorded in macaque IT, and performs well in capturing human behavioural similarity judgements (Kar et al., 2019; Kubilius et al., 2019). Nodal activations for each area of CORnet-S were extracted from the area's convolutional layer. From these, we calculated representational dissimilarity matrices (RDMs) that quantify how similar or different the activations of the layer were between all the images.

In addition, we have expanded our initial RSA analyses of the MEG data to test for semantic effects in the context of a range of popular ANNs – CORnet-S, CORnet-RT, Alexnet, Resnet50 and VGG19. These results show that across this range of ANNs, we see similar semantic effects. These results have been included as Supplementary results (Figure S2):

Figure S2. RSA results for the semantic feature model after different ANNs have been partialled out. Shaded areas show standard error of the mean and dotted line shows the non-partial RSA results for the semantic feature model.

- The semantic feature model should be explained in Intro/Results, and possibly a figure. (The sentence “The semantic model is created using semantic features” caption Fig 1, is not so informative).

We have now added a description of the semantic feature approach to the beginning of the results section, to complement the descriptions in the methods. This addition can be seen on page 4/5, and below:

Modelling the semantics of objects requires an approach that defines semantic similarities and differences between individual concepts. This was achieved by modelling the relationships between concepts according to the semantic features associated with each individual concept (e.g. has legs, has stripes, lives in India, is dangerous for a tiger). Utilising a large-scale property-norming study (Devereux et al., 2014), the similarity between object concepts can be calculated based on the amount of features two concepts share, resulting in a semantic feature RDM. Similarity therefore captures both superordinate category structure (as objects from the same category will have many overlapping features) and additional within-category individuation (as each member of a category will have a unique set of features). These RDMs from CORnet-S and semantic features were then tested against RDMs derived from fMRI and MEG.

- ‘RSA model fit’ is not the right term to use, what is reported are (partial) correlations; the use of ‘model fit’ suggests some kind of optimisation procedure as is common in encoding model approaches that use cross-validation or testing on on held out test data, all of which is not applied here as far as I can tell.

This has now been replaced with RSA model correlations.

fMRI RSA searchlight section:

- The text says the analysis ‘replicates Clarke & Tyler 2014’, but the analysis is based on the same data, so it seems incorrect to call it a replication. It would be good provide more detail on how the current analyses or models used are different from the original analyses in the paper the data is from.

Yes we agree this was not correct. We have now made this explicit, and highlighted the point of including this analysis and how it relates to the original analysis. This can be seen on page 6 and below:

This pattern is similar to our other analyses of semantics using the same dataset that characterised visual properties using different ANNs (AlexNet and HMax) and controlled for different aspects of semantic similarity structure (Clarke & Tyler, 2014; Devereux et al., 2018). Our fMRI analysis here provides a comparison to our MEG source localised RSA effects (see Figure 5) whilst also testing if semantic feature effects remain after controlling for all layers of CORnet-S.

- The section also lacks a clear conclusion. Why is this fMRI analysis included in the first place? Its interpretation (based on the information provided in the Introduction) is not clear.

Thank you for pointing this out, and we have now added a conclusion and justification for this, as can be seen from the response above.

- Page 4, what is 'voxel PS'?

This has now been replaced with voxel pattern similarity.

- Figure 2: showing one hemisphere upside down is confusing imo. Caption B refers to 'model RDM in A', but A shows no RDMs.

As the upside down brains can be potentially confusing, we have now switched these. For panel B, we intended this to refer to the RSA effects for each model, rather than the model itself, and see why this may have been unclear. This has now been adjusted as follows:

Figure 2. fMRI searchlight results. A) Maps show significant relationship between each model RDM and voxel patterns, voxelwise $p < 0.001$, cluster $p < 0.05$. B) Maps showing which model RDM had the strongest effect size at each searchlight centre voxel. C) RSA effects of the semantic feature RDM partialling out effects of all CORnet layers.

MEG section:

- Fig 3:

A. As in Fig 2, it would be helpful to also show the 'regular' RSA correlations (not just the partial ones). Based on the current plot in 2A, it seems strange that there is so little unique correlation of CorNet-V2, but this could be because it shares a lot of variance with CorNet-V1, for example.

Yes, the small CORnet-V2 correlations come from the high level of correlation it has with other layers. We have now added the non-partial analysis to the supplementary results and referenced this in the text (Figure S1).

Figure S1. RSA results for the MEG sensor array using non-partial correlation RSA analysis for each model RDM over time. Shaded areas show standard error of the mean. Solid bars show time periods of significant effects.

B. I'm not so sure what to make of these two plots, which seem to show the same data but referenced to another different RDM (IT vs semantic): is the point that the difference between CorNet-IT and semantic features in terms of onset is not significantly different? It would be helpful to indicate that explicitly (e.g., with a 'significance star'). And why do the distributions look different in shape when taking the CorNet-IT vs. the semantic model as reference?

The aim of the plots is to illustrate the reliability of the peak differences for the different resamples of the data. These are used for the 95% CIs and the important point is that the distributions are pushed away from zero. We've now emphasised this in the text and figure legend, and added in all the 95% CIs in the text. We've also added a line at zero, to emphasise where differences to this line exist, on page 7.

An analysis of the latency of the peak RSA effect sizes was conducted by calculating the 95% confidence intervals of the differences in RSA peak latencies between model RDMs (Figure 3B). This further revealed that the semantic feature model peaked later compared to the low and mid-level CORnet layers (V1 95% CIs of peak difference [172–288 ms], V2 95% CIs [146–262 ms] and V4 95% CIs [174–292 ms]), indicated by the 95% confidence intervals not including zero, but not CORnet-IT (95% CIs [-4–124 ms]), while CORnet-IT peaked later compared to the low and mid-level CORnet layers (V1 95% CIs [156–198 ms], V2 95% CIs [128–178 ms] and V4 95% CIs [156–202 ms]).

- Fig 4: Again here the text claims to show some significant effects in panel C but this is not clearly indicated in the figure. The text also doesn't report a p-value (or percentage of the CI distribution) for the significant differences.

We've now amended the text to refer to reliable differences and reported the 95% CIs in the text too, along with a statement for how we assess if the difference is reliable or not (see previous response). We've used similar conventions to Teichmann et al., (<https://doi.org/10.1523/JNEUROSCI.0158-20.2020>) for reporting peak differences based on 95% CIs, which typically don't give a p-value for each difference.

- Page 7 'concomitant feedback signals [...] pointing to the importance of recurrent activity': these results don't show the recurrent activity is important, they just show it is present. To draw a conclusion about importance, it would be necessary to show that eliminating this recurrent component leads to impaired object recognition, for example.

We have now adjusted this to reflect the potential importance of recurrent activity, as can be seen below and on page 9:

Interestingly, concomitant feedback signals were associated with higher-level visual and semantic properties pointing to the potential importance of recurrent activity during object recognition.

- Fig 5: It's confusing to discuss panel B before panel A. Panel C and D are not referenced in the text.

The figure has now been rearranged to match the order they appear in the text, and reference to all panels has been added.

Figure 5. Searchlight RSA on source-localised MEG signals. A) Semantic feature effects were seen in four spatio-temporal clusters across bilateral pVTC, right ATL and left frontal/ATL. B) Onset time of the semantic effects at each vertex and C) time of peak RSA effect size at each vertex. D) Maps showing significant effects of semantic features (partialling out all effects of the ANN layers) over time.

- Page 7 '[...] accessing semantic information about objects': I'm not sure if it is valid to conclude that these model correlations mean that semantic information was 'accessed'; the subjects were just naming the objects, perhaps this data is suggestive that they activated some semantic properties while doing so, but the process that underlies this is not clear from these data.

This has been adjusted as follows and on page 9:

This shows semantic object processing engages a network of bilateral pVTC, ATL and the left ventral PFC within the first 600 ms of an object appearing.

- Page 8 'with spatial distribution of semantic effects closely overlapping those revealed through fMRI RSA': it's quite hard to assess if this is accurate from the few (small) brain pictures shown. A claim like this should be supported by a quantitative analysis of the overlap.

A quantitative analysis was considered, however this is problematic as the MEG searchlight analysis is conducted on the cortical surface and the fMRI searchlight analysis is conducted in voxel space. We can convert the data between voxel and vertex spaces, but data is lost and not all voxels can be represented on vertices. Therefore we have rephrased the statement to draw attention to the broad regions both analysis pick out. This can be seen on page 9 and below:

These effects for semantics are seen over and above those explained by the CORnet ANN of vision, with the spatial distribution of semantic effects including ventral temporal, anterior temporal and lateral occipital regions that were also present in the fMRI RSA searchlight results.

Multivariate RCA analysis (Page 8 & Figure 6):

- I had trouble understand this analysis. In my understanding whether an MEG-RSA correlation could be considered 'feedforward' or 'feedback' is based on whether it is partialing out the time-points preceding current time t anterior vs. posterior sensors (Fig 1). How is feedforward determined using these clusters?

We have now elaborated on the RCA methods in various places to make the approach clearer. RCA is now more extensively introduced in the introduction on page 3:

To address this, we use a recently developed method for model-based representational connectivity analysis (RCA; Karimi-Rouzbahani et al., 2021, 2022), which builds on prior examples of understanding connectivity using representational similarity approaches for fMRI (Anzellotti & Coutanche, 2018; Clarke et al., 2022; Kriegeskorte, 2008; Pillet et al., 2020). In the case of time-resolved data, RCA can show how activity in one region contributes to information-specific activity in another region at a later point in time, with the potential to reveal how feedforward and feedback signals relate to visual and semantic information. This follows the recent development of multivariate informational connectivity approaches (Anzellotti & Coutanche, 2018; Basti et al., 2020; Goddard et al., 2016; Rahimi, Jackson, et al., 2022), which seek to provide information about the timing and direction of between-region connectivity, alongside pointing towards the representational content connectivity can support. To infer timing and direction, RCA uses similar principles to Granger Causality, in that it evaluates whether past information in one region can help explain the current representational patterns in another region. While this kind of approach has been used in a number of studies to evaluate if there is shared information between regions over time (e.g. Goddard et al., 2022; Kietzmann et al., 2019; Lyu et al., 2019), the RCA approach we employ here goes one step further by specifying the precise nature of that shared information (Karimi-Rouzbahani et al., 2021, 2022) – namely, is connectivity helping shape visual or semantic representations.

And described more in regards to the MEG sensor analysis on page 7/8:

The analysis aims to test what influence past representational similarity in a source region (e.g. posterior sensors) has on future RSA effects of semantics in a target region (e.g. anterior sensors). For example, if there is feedforward connectivity whereby patterns in posterior regions influence patterns in anterior regions that relate to semantics, then those patterns in posterior regions should help explain variance in the anterior patterns. We can assess this by calculating two region-specific, time-lagged RSA time-courses and finding the difference. First, we calculate RSA effects in the target region for a model, and second we calculate RSA effects in the target region for a model while controlling for past neural similarity effects in the source region. If the neural patterns in the source region help

explain something about the models effects in the target region, then the RSA effects will be reduced. This reduction indicates that past responses in the source region influence the current similarity in the target region, and do so in a way specific to that model RDM being tested.

And in Figure 4A:

Figure 4. Illustration and results of the MEG sensor-level Representational connectivity analysis (RCA). A) Illustration of the calculation of feedforward information flow (feedforward connectivity at each timepoint) between anterior and posterior regions at the MEG sensor-level, as introduced by Karimi-Rouzbahani et al. (2021). i) RDMs are created from posterior and posterior sensors. Feedforward flow at each timepoint is formalised as the contribution of the earlier posterior RDM (t-30m) to the current model-anterior RDM correlation (t). This is calculated as the difference between the anterior-model RDM correlation and the anterior-model RDM correlation where the posterior RDM is partialled out. Feedback information flow is formalised as the contribution of the earlier anterior RDM (t-30) to the current model-posterior RDM correlation (t). ii) In the partial RCA, the contribution of other model RDMs is also partialled out in the calculation of both RSA timecourses. B) Feedforward RCA effects for each model RDM. C) Feedback effects of the model RDMs. Shaded areas show standard error of the mean. Solid bars show time periods of significant effects. D) Swarmplots showing the differences in peak RCA latency between model RDMs. Distributions display resamples of the data which were used to generate 95% CIs for the differences in peak latencies.

And we highlight how this is extended for the analysis of the source-localised data from the three clusters, on page 11:

We created our network to consist of three ROIs based on the significant RSA clusters, being bilateral pVTC (clusters 1 and 2), right ATL (cluster 3), and left vPFC/ATL (cluster 4). Any areas of spatial overlap between the clusters were excluded from all ROIs. Using these regions, we aimed to test the influence of past representational similarity in a source region on future RSA effects of semantics in a target region. However, we also want to control for the past representational similarity from other regions in the network, as well as CORnet. To do this we used a multivariate RCA measure which tests whether past similarity in the source region helps explain future semantic effects in the target region while also controlling for the other region in our network.

And in Figure 6A:

Figure 6. Illustration and results of RCA for source localised MEG signals. A) RCA analysis applied to ROI clusters. Feedforward effects in the source-level RCA is formalised in the same way as in the sensor-level analysis, and is based on the difference of two partial correlations. (i) The first measures the relationship between the target neural RDM and the semantic feature RDM, while controlling for other model RDMs and past RDMs from control regions. (ii) The second correlation measures the relationship between the target RDM and the semantic feature RDM, while controlling for the same other factors in addition to also removing the effects of past similarities in the source region. (iii) Example time-courses of these two partial correlations. A reduced correlation in the second partial correlation indicates the contribution of the source region to the target regions RSA effect. (iv) Subtracting the second correlation from the first is the RCA measure. B) Effects between the pVTC and right ATL. Solid line shows feedforward effects (pVTC → rATL) and line with circles shows feedback effects (rATL → pVTC). C) Feedforward (pVTC → PFC/ATL) and Feedback (PFC/ATL → pVTC) RCA effects between pVTC and the left PFC/ATL. D) Feedforward (rATL → PFC/ATL) and Feedback (PFC/ATL → rATL) RCA effects between right ATL and left PFC/ATL. Shaded areas show standard error of the mean. Solid bars show time periods of significant effects.

On this last part, whether something is feedforward or feedback relates to the posterior to anterior axis, where pVTC is the most posterior area, then right ATL, and left ATL/PFC is the most anterior.

- While the results seem somewhat consistent across panels, here I also started to feel a bit uncomfortable with the extremely small correlation values on the y-axis. I understand these are partial correlations but still, how meaningful are effects of this size?

The RCA effects sizes appear small with respect to typical RSA analyses (which also tend to have small correlation values in the range $\rho = 0.05-0.2$), and this is because RCA values are the difference in two RSA calculations (both of which are partial correlations). It is now made more explicit in the manuscript that RCA is the difference of two RSA effects, as shown in the changes referenced above. Additionally, while the RCA values might be small differences between correlations, our new analysis relating behaviour to RCA suggests that these are meaningful effects in helping shape behaviour, as discussed above.

- There also seems to be potential circularity in the analysis, given that the clusters are first identified based on whether they show a unique partial correlation with semantic vs. ANN features, and then evaluated again for significance on this correlation in the plots in Fig. 6.

Thank you for raising this, which is hopefully clarified now we have more explicitly described the RCA methods. Specifically relating to this point, we do not see a circularity in the analyses, as the RSA analyses shows the presence of semantic effects, while the RCA analysis is based on the difference of two RSA calculations both of which include the semantic model.

Methods:

- The model specification should include what data CorNet was trained on.

This has now been included on page 17 and below:

Visual model RDMs were created using CORnet-S, a four-layer recurrent DNN for core object recognition which was modelled on the anatomy of the primate ventral visual stream, encompassing areas V1, V2, V4 and IT (Kubilius et al., 2019) and trained on ImageNet.

- MEG data: How many images/trials in total? How many repeats?

This is included on page 16 and below:

All participants performed a basic-level naming task during which they were asked to identify 302 common objects from 11 superordinate categories, comprising of both living and non-living things. All objects were shown to participants in a pseudo-randomised order and presented in colour on a white background. A trial began with a fixation cross for 500 ms, before the object was presented for 500ms, after which followed a blank screen for a random interval between 2400ms and 2700ms. Each object was only shown once.

- Peak latency analysis: I don't understand how the n-leave-out is used in the determining the significance. The paragraph basically describes a permutation analysis, but doesn't talk about how the left-out data is used in this context.

The peak latency analysis uses confidence intervals to determine if the differences in the peak latencies between two model RDMs is reliably non-zero. To do this, we need to generate a large number of resamples of the data which are used to estimate the range of peak differences likely if we repeated the experiment. To obtain these resamples, we used a leave-n-out approach, rather than bootstrapping with replacement, to generate 31,465 resamples of the data, with the left out items not being used for anything (so this isn't a leave-n-out cross validation approach). This gives us a range of peak difference that we might see if the experiment was repeated, from which the confidence intervals are derived. We want to know if the range contains zero – i.e. if zero falls within our CIs, then we have no evidence that the differences in the peaks is meaningful. If the CIs do not contain or cross zero, then we have evidence that there is a reliable difference in the peak latencies. This has now been expanded in the results (see response above) and on page 18, included below:

In order to assess the statistical significance of the difference in peak latencies, we used a leave-n-out jackknife approach, with n set to 4 (12% of the data; highly similar results are seen at n=3 and n=5). The leave-n-out approach was used in place of bootstrapping to create a large number of resamples of the data from which confidence intervals for the differences in peak latencies can be calculated. For each of the possible 31,465 unique resamples of the data, group average RSA time-courses were calculated and the timepoint of the maximum effect was extracted. This created a distribution of peak times for each model RDM. The distribution of peak was then used to define 95% confidence intervals (CIs) of the peak, and distribution of peak differences to define 95% CIs of pairwise differences. The null

hypothesis was rejected if the 95% CIs of the pairwise differences did not include zero (Mohsenzadeh et al., 2019).

- What software was used for these analyses? Matlab, Python, any pre-existing toolboxes? Is the data publicly available?

For the fMRI analysis, Matlab was used along with the RSA toolbox (Nili et al., 2014) hosted here: <https://www.mrc-cbu.cam.ac.uk/methods-and-resources/toolboxes/license/>. For the MEG analysis, Matlab was also used. As described, SPM12 was used for the preprocessing of the data. RSA analyses were conducted using a custom toolbox, and source-level searchlight analysis using the toolbox here (<https://github.com/AlexDClarke>).

We have now added a data availability statement and code availability statement to the manuscript on page 20/21, and as follows:

Data availability statement

The data used in this research was obtained from different experiments with different availabilities. The MEG data collected as part of the Cambridge Centre for Ageing and Neuroscience was part of stage III and is available upon requested (see <https://camcan-archive.mrc-cbu.cam.ac.uk/dataaccess/>). The remaining MEG data is found here <https://osf.io/2uqf4/>, and fMRI data here <https://osf.io/e2s59/>.

Code availability

The custom code used for the MEG RSA analysis is available here - https://github.com/AlexDClarke/MEG_RSA_VonSeth.

- The searchlight RSA section doesn't explicitly say this analyses was done in the source-reconstructed data. The source reconstruction itself is only very minimally described in MEG preprocessing section, I almost missed it.

This has now been included on page 18 and below:

The relationship between the representational geometries of the cognitive RDMs and of the MEG signals was tested using spatiotemporal searchlight RSA (Su et al., 2012) of the source localised MEG signals.

- fMRI experiment description: seems more logical to start Methods with this since this is first reported in results. I found it hard to follow the details of the methods with the minimal information provided. For example, what is 'the grey matter mask' that was used to mask the t-maps; how was it obtained? Why were 'unnormalised EPI images' used to create the t-maps; unnormalized in what way? Unlike the MEG section, the fMRI section also lacks detail on how the RDM-correlations were obtained.

We have now expanded this section, but still limit our descriptions to the essential information and refer to the original paper. The expanded descriptions can be seen on page 19-20, and below:

fMRI scanning and Preprocessing

Participants were scanned at the MRC Cognition and Brain Sciences Unit, Cambridge, in a Siemens 3-T Tim Trio MRI scanner (Siemens Medical Solutions, Camberley, UK). There were 3 functional scanning sessions using standard gradient-echo echoplanar imaging (EPI) sequences with 3 x 3 x 3 mm voxel

size. Prior to functional scanning, a high-resolution structural MRI image was collected using an MP-RAGE sequence with 1 mm isotropic resolution. Preprocessing of the functional data consisted of slice-time correction and the spatial realignment of the functional images only using SPM8 (Wellcome Institute of Cognitive Neurology, London, UK). The un-normalised and non-smoothed EPI images were entered into a general linear model to obtain a single t-statistic image for each concept picture based on all 6 repetitions. Activity in these t-maps was then restricted to voxels identified as grey matter, as identified from the T1 images, and used in subsequent representational similarity analysis.

fMRI searchlight RSA

An RSA searchlight mapping procedure (Kriegeskorte et al., 2006) was implemented using the RSA toolbox (Nili et al., 2014) in Matlab to determine if object dissimilarity predicted by the visual and semantic models was significantly related to dissimilarity defined by local fMRI activity patterns. At each voxel, object activation values from grey matter voxels within a spherical searchlight (radius 7 mm) were extracted to calculate distances between all objects (using $1 - \text{Pearson correlation}$) creating an object dissimilarity matrix based on that searchlight. This fMRI RDM is then compared to each model RDM (using Spearman's rank correlation) and mapped back to the voxel at the centre of the searchlight.

For group random-effects analyses, the Spearman's correlation maps for each participant were Fisher-transformed, normalized to standard MNI space, and spatially smoothed with a 6mm FWHM Gaussian kernel. Maps were entered into a random effects analysis (RFX) in SPM12 where each voxel was tested using a one-sampled t-test against zero. Voxelwise multiple comparisons correction was applied through a voxelwise threshold of $p < 0.001$ and FWE-cluster $p < 0.05$.

- Sensor RCA section 'in contrast to RSA analysis, upper triangle of each RDM was converted to an RDV'; usually in RSA also only the upper (or lower) triangle is used when comparing RDMs, this is important because including the whole matrix and/or the diagonal can inflate correlations (see Ritchie et al., 2017).

Sorry for this confusion. All the analyses, both RSA and RCA, MEG and fMRI, only use the upper triangle of each RDM. The whole matrix is never used. We have now made this clearer in the text and in the figures.

References

Anzellotti, S., & Coutanche, M. N. (2018). Beyond Functional Connectivity: Investigating Networks of Multivariate Representations. *Trends in Cognitive Sciences*, 22(3), Article 3.
<https://doi.org/10.1016/j.tics.2017.12.002>

Clarke, A., Crivelli-Decker, J., & Ranganath, C. (2022). Contextual Expectations Shape Cortical Reinstatement of Sensory Representations. *Journal of Neuroscience*, 42(30), Article 30.
<https://doi.org/10.1523/JNEUROSCI.2045-21.2022>

Goddard, E., Carlson, T. A., Dermody, N., & Woolgar, A. (2016). Representational dynamics of object recognition: Feedforward and feedback information flows. *NeuroImage*, 128, 385–397.
<https://doi.org/10.1016/j.neuroimage.2016.01.006>

- Goddard, E., Contini, E. W., & Irish, M. (2022). Exploring Information Flow from Posteromedial Cortex during Visuospatial Working Memory: A Magnetoencephalography Study. *Journal of Neuroscience*, 42(30), 5944–5955. <https://doi.org/10.1523/JNEUROSCI.2129-21.2022>
- Karimi-Rouzbahani, H., Vahab, E., Ebrahimpour, R., & Menhaj, M. B. (2019). Spatiotemporal analysis of category and target-related information processing in the brain during object detection. *Behavioural brain research*, 362, 224-239.
- Karimi-Rouzbahani, H., Ramezani, F., Woolgar, A., Rich, A., & Ghodrati, M. (2021). Perceptual difficulty modulates the direction of information flow in familiar face recognition. *NeuroImage*, 233, 117896. <https://doi.org/10.1016/j.neuroimage.2021.117896>
- Karimi-Rouzbahani, H., Woolgar, A., Henson, R., & Nili, H. (2022). Caveats and Nuances of Model-Based and Model-Free Representational Connectivity Analysis. *Frontiers in Neuroscience*, 16. <https://www.frontiersin.org/articles/10.3389/fnins.2022.755988>
- Kietzmann, T. C., Spoerer, C. J., Sörensen, L. K. A., Cichy, R. M., Hauk, O., & Kriegeskorte, N. (2019). Recurrence is required to capture the representational dynamics of the human visual system. *Proceedings of the National Academy of Sciences*, 116(43), 21854–21863. <https://doi.org/10.1073/pnas.1905544116>
- Kriegeskorte, N. (2008). Representational similarity analysis – connecting the branches of systems neuroscience. *Frontiers in Systems Neuroscience*. <https://doi.org/10.3389/neuro.06.004.2008>
- Lyu, B., Choi, H. S., Marslen-Wilson, W. D., Clarke, A., Randall, B., & Tyler, L. K. (2019). Neural dynamics of semantic composition. *Proceedings of the National Academy of Sciences*, 116(42), 21318–21327. <https://doi.org/10.1073/pnas.1903402116>
- Nili, H., Wingfield, C., Walther, A., Su, L., Marslen-Wilson, W., & Kriegeskorte, N. (2014). A toolbox for representational similarity analysis. *PLoS Comput Biol*, 10(4), Article 4. <https://doi.org/10.1371/journal.pcbi.1003553>
- Pillet, I., Op de Beeck, H., & Lee Masson, H. (2020). A Comparison of Functional Networks Derived From Representational Similarity, Functional Connectivity, and Univariate Analyses. *Frontiers in Neuroscience*, 0. <https://doi.org/10.3389/fnins.2019.01348>
- Teichmann, L., Quek, G. L., Robinson, A. K., Grootswagers, T., Carlson, T. A., & Rich, A. N. (2020). The influence of object-color knowledge on emerging object representations in the brain. *Journal of Neuroscience*, 40(35), 6779-6789.

REVIEWERS' COMMENTS:

Reviewer #1 (Remarks to the Author):

The authors addressed the concerns the reviewers have raised and the manuscript at the current state is presented much stronger, especially with the addition of the behavioral relevance of the recurrent activity part.

Also the authors made it clear how their semantic relational model is different from visuo-semantic models used in previous studies. The author's point on potentially creating an artificial network with visual, high-level visuo-semantic, and semantic relational representations is insightful.

I have some minor comments about the structure and grammar of the abstract. I've edited some parts below:

Visual object recognition has been Traditionally conceptualised as a sequential, predominantly feedforward process supported by the ventral visual pathway. It has been modeled using mostly feedforward artificial convolutional neural networks (CNNs), which can achieve human-level classification on some image-labelling tasks. However, it remains unclear whether these computational models of vision, mostly with feedforward connection only, alone can accurately capture the evolving spatiotemporal neural dynamics. Here, we probe these dynamics using a combination of representational similarity and connectivity analyses of fMRI and MEG data recorded during the recognition of familiar, unambiguous objects from a wide range of categories. Modelling the visual and semantic properties of our stimuli using an artificial neural network as well as a semantic feature model, we find that unique aspects of the neural architecture and connectivity dynamics relate to visual and semantic object properties. Critically, we show that recurrent processing between anterior and posterior ventral temporal cortex relates to higher-level visual properties prior to semantic object properties, in addition to semantic-related feedback from the frontal lobe to the ventral temporal lobe between 250 and 500ms after stimulus onset. These results demonstrate the distinct contributions made by semantic object properties in explaining neural activity and connectivity, highlighting it as a core part of object recognition not fully accounted for by current state of the art biologically inspired ANNs.

It would be helpful if the authors can highlight the importance of including semantic model. For example, "visual object recognition naturally engages our conceptual semantic knowledge about object categories...The inclusion of semantic model may account for the dynamic nature of the visual object recognition in the brain."

Reviewer #2 (Remarks to the Author):

The authors have successfully addressed my concerns. I am happy with this manuscript to move forward.

REVIEWERS' COMMENTS:

Reviewer #1 (Remarks to the Author):

The authors addressed the concerns the reviewers have raised and the manuscript at the current state is presented much stronger, especially with the addition of the behavioral relevance of the recurrent activity part.

Also the authors made it clear how their semantic relational model is different from visuo-semantic models used in previous studies. The author's point on potentially creating an artificial network with visual, high-level visuo-semantic, and semantic relational representations is insightful.

I have some minor comments about the structure and grammar of the abstract. I've edited some parts below:

Visual object recognition has been traditionally conceptualised as a sequential, predominantly feedforward process supported by the ventral visual pathway. It has been modeled using mostly feedforward artificial convolutional neural networks (CNNs), which can achieve human-level classification on some image-labelling tasks. However, it remains unclear whether these computational models of vision, mostly with feedforward connection only, alone can accurately capture the evolving spatiotemporal neural dynamics. Here, we probe these dynamics using a combination of representational similarity and connectivity analyses of fMRI and MEG data recorded during the recognition of familiar, unambiguous objects from a wide range of categories. Modelling the visual and semantic properties of our stimuli using an artificial neural network as well as a semantic feature model, we find that unique aspects of the neural architecture and connectivity dynamics relate to visual and semantic object properties. Critically, we show that recurrent processing between anterior and posterior ventral temporal cortex relates to higher-level visual properties prior to semantic object properties, in addition to semantic-related feedback from the frontal lobe to the ventral temporal lobe between 250 and 500ms after stimulus onset. These results demonstrate the distinct contributions made by semantic object properties in explaining neural activity and connectivity, highlighting it as a core part of object recognition not fully accounted for by current state of the art biologically inspired ANNs.

Response:

Thank you for these suggestions, some of which we have adopted, and others not for space constraints. A modified version of the abstract is included below:

Visual object recognition has been traditionally conceptualised as a predominantly feedforward process through the ventral visual pathway. While feedforward artificial neural networks (ANNs) can achieve human-level classification on some image-labelling tasks, it's unclear whether computational models of vision alone can accurately capture the evolving spatiotemporal neural dynamics. Here, we probe these dynamics using a combination of representational similarity and connectivity analyses of fMRI and MEG data recorded during the recognition of familiar, unambiguous objects. Modelling the visual and semantic properties of our stimuli using an artificial neural network as well as a semantic feature model, we find that unique aspects of the neural architecture and connectivity dynamics relate to visual and semantic object properties. Critically, we show that recurrent processing between

anterior and posterior ventral temporal cortex relates to higher-level visual properties prior to semantic object properties, in addition to semantic-related feedback from the frontal lobe to the ventral temporal lobe between 250 and 500ms after stimulus onset. These results demonstrate the distinct contributions made by semantic object properties in explaining neural activity and connectivity, highlighting it as a core part of object recognition not fully accounted for by current biologically inspired neural networks.

It would be helpful if the authors can highlight the importance of including semantic model. For example, "visual object recognition naturally engages our conceptual semantic knowledge about object categories...The inclusion of semantic model may account for the dynamic nature of the visual object recognition in the brain."

Response:

A statement to this effect has been added near the beginning of the introduction:

'Successful visual object recognition enables us to see and understand the world around us. It is well established that the ventral visual pathway (VVP) critically supports this process (Clarke & Tyler, 2015; DiCarlo et al., 2012; Kravitz et al., 2013; Ungerleider & Mishkin, 1982), with activity within the first few hundred milliseconds supporting the recognition of the visual input (Bankson et al., 2018; T. Carlson et al., 2013; Chen et al., 2016; Cichy et al., 2014, 2016; Clarke et al., 2013, 2015, 2018; Giari et al., 2020; Isik et al., 2014; Rupp et al., 2017). Our increasingly detailed computational, cognitive and translational accounts of the spatiotemporal processes that underly recognition allow us to form an increasingly specific understanding of what kinds of information may be linked to the evolving object representations, and how they are spatially and temporally distributed. Visual object recognition intrinsically involves accessing semantic knowledge about objects, highlighting the necessity to consider how visual and semantic properties account for the neural dynamics during recognition. While prior studies have shed some light on how semantic object processing relates to, and is distinct from, processing the physical visual attributes of the image (Bankson et al., 2018; Clarke et al., 2013, 2015, 2018; Jozwik et al., 2023; Rupp et al., 2017), in this study we look to probe the evolving nature of object representations in a new domain, asking how dynamic connectivity patterns relate to the visual and semantic aspects of objects. Using a representational connectivity approach (Karimi-Rouzbahani et al., 2022) will allow us to reveal the core object properties that feedforward and feedback connectivity relate to, helping to shape cognitive accounts of how we understand the meaning of what we see.'

Reviewer #2 (Remarks to the Author):

The authors have successfully addressed my concerns. I am happy with this manuscript to move forward.